# NESTEROV ACCELERATION IN BENIGNLY NON-CONVEX LANDSCAPES

**Kanan Gupta, Stephan Wojtowytsh**
Department of Mathematics, University of Pittsburgh
kanan.g@pitt.edu, s.woj@pitt.edu

## ABSTRACT

While momentum-based optimization algorithms are commonly used in the notoriously non-convex optimization problems of deep learning, their analysis has historically been restricted to the convex and strongly convex setting. In this article, we partially close this gap between theory and practice and demonstrate that virtually identical guarantees can be obtained in optimization problems with a 'benign' non-convexity. We show that these weaker geometric assumptions are well justified in overparametrized deep learning, at least locally. Variations of this result are obtained for a continuous time model of Nesterov's accelerated gradient descent algorithm (NAG), the classical discrete time version of NAG, and versions of NAG with stochastic gradient estimates with purely additive noise and with noise that exhibits both additive and multiplicative scaling.

## 1 INTRODUCTION

Accelerated first order methods of optimization are the backbone of modern deep learning. So far, theoretical guarantees that momentum-based methods accelerate over memory-less gradient-based methods have been limited to the setting of convex objective functions. Indeed, recent work of Yue et al. (2023) shows that the assumption of convexity cannot be weakened as far as, for instance, the Polyak-Lojasiewicz (PL) condition $\|\nabla f\|^2 \geq 2\mu\left(f - \inf f\right)$, which has been used to great success in the study of gradient descent algorithms *without* momentum for instance by Karimi et al. (2016).

Optimization problems in deep learning are notoriously non-convex. Initial theoretical efforts focused on approximating the training of very wide neural networks by the parameter optimization in a related linear model: The neural tangent kernel (NTK). Jacot et al. (2018); E et al. (2019) show that for randomly initialized parameters, gradient flow and gradient descent trajectories remain uniformly close to those which are optimized by the linearization of the neural network around the law of its initialization. This analysis was extended to momentum-based optimization by Liu et al. (2022).

Recall that a $C^2$-smooth function $f$ is convex if and only if its Hessian, $D^2 f$, is positive semi-definite, i.e. has only non-negative eigenvalues. Strictly negative (but small) eigenvalues of the Hessian of the loss function have been observed close to the set of global minimizers experimentally by Sagun et al. (2017; 2018); Alain et al. (2018) and their presence has been explained theoretically by Wojtowytsch (2023). This poses questions about the use of momentum-based optimizers such as SGD with (heavy ball or Nesterov) momentum or Adam in the training of deep neural networks. In this work, we show that acceleration can be guaranteed for Nesterov's method under much weaker geometric assumptions than (strong) convexity, in particular for certain objective functions that have non-unique and non-isolated minimizers and whose Hessian may have negative eigenvalues up to a certain size.

In the remainder of this section, we briefly review how our work fits into the literature. In Section 2, we precisely state the assumptions under which we prove convergence at an accelerated rate and discuss how our work connects to optimization in deep learning. Our main results are presented in Section 3, both in discrete and continuous time. Some technical details are postponed to the appendix.

### 1.1 PREVIOUS WORK

Gradient-based optimization was first proposed by Cauchy et al. (1847) in form of the gradient descent algorithm. Over a century later, momentum-based 'accelerated' algorithms were introduced

by Hestenes & Stiefel (1952) for convex quadratic functions and by Nesterov (1983) for general smooth and convex objective functions. Nesterov's work was generalized to non-smooth convex optimization by Beck & Teboulle (2009) and to stochastic smooth convex optimization among others by Nemirovski et al. (2009); Shamir & Zhang (2013); Jain et al. (2019); Laborde & Oberman (2020) for additive noise and by Liu & Belkin (2018); Even et al. (2021); Vaswani et al. (2019); Gupta et al. (2024) for multiplicatively scaling noise. See also (Ghadimi & Lan, 2012; 2013) for more information on accelerated stochastic gradient methods. While the heavy ball method is used extensively in deep learning, Lessard et al. (2016); Goujaud et al. (2023) prove that it does not generally achieve accelerated convergence for smooth strongly convex functions and may even diverge – see however (Kassing & Weissmann, 2024) for positive results under stronger smoothness assumptions

Accelerated gradient methods have been studied e.g. by Josz et al. (2023) under much weaker regularity conditions and weaker geometric conditions than (strong) convexity, namely the Kurdyka-Lojasiewicz (KL) condition. Under those weaker assumptions, it is at best possible to prove convergence to a local minimizer at a non-accelerated rate: Under the (comparatively weak) Polyak-Lojasiewicz (PL) condition, a special case of the KL condition, Yue et al. (2023) show that it is not possible to obtain an accelerated rate of convergence. A slower linear rate of convergence is established by Apidopoulos et al. (2022) in continuous time under the assumption that the objective function $f$ satisfies has an $L$-Lipschitz continuous gradient and satisfies the PL-inequality $2\mu(f - \inf f) \le \|\nabla f\|^2$. The rate of convergence is

$$\sqrt{\mu}\left(\sqrt{L/\mu} - \sqrt{L/\mu - 1}\right) = \sqrt{L}\left(1 - \sqrt{1 - \frac{\mu}{L}}\right) \approx \sqrt{L} \cdot \frac{\mu}{2L} = \frac{\mu}{2\sqrt{L}}.$$

A stable time-discretization can generally be attained with effective step-size $1/\sqrt{L}$ for momentum methods (see Su et al., 2016, Section 2), suggesting convergence at the non-accelerated linear rate $(1 - \mu/2L)^k$ in discrete time. To the best of our knowledge, no proof has been given yet.

There have been several efforts to find a reasonable relaxation of convexity for which accelerated convergence can still be achieved. Hinder et al. (2020); Fu et al. (2023); Wang & Wibisono (2023); Guminov et al. (2023) consider acceleration under the weaker condition that the objective function is $\gamma$-quasar or $(\gamma, \mu)$-strongly quasar-convex, i.e. the inequality

$$\langle \nabla f(x), x - x^* \rangle \ge \gamma\left(f(x) - f(x^*) + \frac{\mu}{2}\|x - x^*\|^2\right)$$

holds for any $x \in \mathbb{R}^d$ and *any* minimizer $x^*$ of $f$. Compared to (strong) convexity, it relaxes the condition in two ways: It only considers pairs $(x, x^*)$ rather than general pairs of points $x, y \in \mathbb{R}^d$, and it introduces a factor $\gamma$ into the inequality which may be strictly smaller than one. Still, it has geometric implications which may be too strong in the context of deep learning: In the strongly quasar-convex case, minimizers are unique, and in the quasar-convex case, sub-level sets are star-shaped with respect to any minimizer $x^*$ since $f(tx + (1 - t)x^*)$ is monotone increasing on $[0, 1]$ (see Lemma 24, Appendix C).

Accelerated rates of convergence were obtained by Necoara et al. (2019) in discrete time and by Aujol et al. (2022) in continuous time under the assumption that the objective function is both convex and quasi-strongly convex, and that it has a unique minimizer. Their results are generalized by Hermant et al. (2024) who allow for non-smooth composite optimization and consider $\gamma \in (0, 1]$ rather than just $\gamma = 1$. Unlike the present work, Hermant et al. (2024) require the uniqueness of minimizers and only study deterministic optimization. Curiously, despite the difference in settings, they independently find the same lower bound on Hessian eigenvalues that we require (compare e.g. Theorem 11 and (Hermant et al., 2024, Theorem 2)). See also (Aujol et al., 2024, Table 1) for an overview of theoretical guarantees of acceleration without strong convexity.

In overparametrized deep learning, the set of minimizers of the loss function is a (generally curved) manifold, and tangential motion to the manifold can have important implications on the implicit bias of an algorithm (Li et al., 2021; Damian et al., 2021). Any notion that takes into account *all* minimizers is quite rigid for such tasks. A more realistic assumption is the 'aiming condition' of Liu et al. (2024) that

$$\langle \nabla f(x), x - \pi(x) \rangle \ge \gamma(f(x) - \min f)$$

where $\pi(x)$ is the closest minimizer to the point $x$. This notion enjoys much greater flexibility in terms of the global geometry of $f$. Liu et al. (2024) investigate the convergence of gradient flows under the aiming condition, but not that of momentum methods.

## 1.2 OUR CONTRIBUTION

Our study can be seen as combining geometric ideas pertaining to $(1, \mu)$-quasar convexity and the aiming condition. Our assumption in (1) is equivalent to quasi-strong convexity (Necoara et al., 2019), but notably we do not make any additional assumptions about the convexity of the objective function or uniqueness of minimizers unlike prior works. Since we focus on the closest minimizer at any point in the trajectory, this presents a unique challenge compared to analyzing the distance from a fixed minimizer. As the current iterate $x_t$ moves, its projection onto the set of minimizers, $\pi(x_t)$, also moves. This requires a modification of the usual Lyapunov function to account for the movement of $\pi(x_t)$ (or the *tangential movement* of $x_t$ parallel to the set of minimizers) as well as the 'drift' in directions where the objective function is negatively curved. To the best of our knowledge, this is the first study which takes into account this tangential movement and obtains accelerated convergence.

In overparametrized learning, the set of minimizers of a loss function is a submanifold of high dimension in a usually much higher-dimensional space. Unless the manifold of minimizers is a linear space, the Hessian of the loss function is geometrically required to have negative eigenvalues in any neighborhood of the set of a minimizer where the manifold is curved. Still, accelerated methods in first order optimization have been found to be highly successful in deep learning. A common heuristic has been that as long as the objective function is convex in the direction towards the set of minimizers, small negative eigenvalues in directions parallel to the set of minimizers can safely be ignored: Tangential drift along the set of minimizers should not affect the decay of the objective function significantly. We prove that this intuition indeed applies in a continuous time model for gradient descent with momentum (Theorems 6 and 8) and for Nesterov's time-stepping scheme (Theorems 11, 13 and 14) in deterministic optimization and stochastic optimization with bounded noise. With 'multiplicative (state-dependent) noise' motivated by overparametrized deep learning, we prove an analogous statement for a modified version of Nesterov's algorithm (Theorem 15).

## 2 SETTING

### 2.1 ASSUMPTIONS

We always make the following assumptions on the regularity and geometry of the function $f$ and its set of minimizers.

1. The objective function $f : \mathbb{R}^d \to \mathbb{R}$ is bounded from below, $C^1$-smooth and its gradient $\nabla f : \mathbb{R}^d \to \mathbb{R}^d$ is locally Lipschitz continuous.

2. The set $\mathcal{M} = \{x \in \mathbb{R}^d : f(x) = \inf_{z \in \mathbb{R}^d} f(z)\}$ of minimizers of $f$ is a (non-empty) $k$-dimensional $C^2$-submanifold of $\mathbb{R}^d$ for $k < d$.

3. There exists an open sub-level set $\mathcal{U}_\alpha = \{x \in \mathbb{R}^d : f(x) < \alpha\}$ for some $\alpha > 0$ such that for every $x \in \mathcal{U}_\alpha$ there exists a unique $z \in \mathcal{M}$ which is closest to $x$. We denote $z$ as $\pi(x)$ and assume that the closest point projection map $\pi : \mathcal{U}_\alpha \to \mathcal{M}$ is $C^1$-smooth.

4. $f$ satisfies the $\mu$-*strong aiming condition* (with respect to the closest minimizer) in $\mathcal{U}_\alpha$, i.e.

$$\nabla f(x) \cdot \big(x - \pi(x)\big) \geq f(x) - f\big(\pi(x)\big) + \frac{\mu}{2} \|x - \pi(x)\|^2 \qquad \forall \, x \in \mathcal{U}_\alpha. \qquad (1)$$

These assumptions are significantly weaker than the assumption of strong convexity, but for instance strong enough to imply a PL inequality with constant $\mu$ (see Appendix C). As illustrated in Section 2.3, they match many geometric features of overparametrized deep learning. For an analysis in discrete time, we will make stronger quantitative assumptions.

### 2.2 SIMPLE EXAMPLES

We give a number of examples which are covered by these assumptions where $f$ is not merely a $\mu$-strongly convex function. The first example illustrates how subtle the interplay between geometric conditions is, even in one dimension.

*Example* 1. For $\varepsilon, R > 0$, consider the function $f(x) = \frac{x^2}{2} + \frac{\varepsilon}{2} x^2 \sin(2R \log(|x|))$. The function $f$ has a unique global minimizer at $x^* = 0$ and its derivative $f'$ is a Lipschitz-continuous function

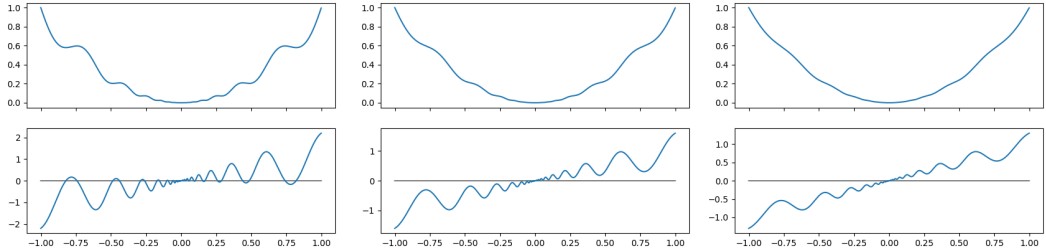

Figure 1: We visualize $f$ from Example 1 in the top row and its derivative in the bottom row with $R = 2$ and $\varepsilon = 0.2$ (left), $\varepsilon = 0.1$ (middle) and $\varepsilon = 0.05$ (right). Left: $f$ has many local minimizers as the derivative crosses 0 an infinite number of times. Middle: $f$ satisfies the PL condition, but not the strong aiming condition. Right: $f$ is strongly aiming (with respect to the unique global minimizer, which implies the PL condition). In all plots, $f$ is non-convex since $f'$ is non-monotone.

with Lipschitz-constant $1 + \varepsilon\sqrt{1 + 5R^2 + 4R^4}$. Furthermore, $f$ has an infinite number of strict local minimizers if $\varepsilon\sqrt{1 + R^2} > 1$, but satisfies favorable geometric properties under stronger assumptions:

|  | PL condition | $\mu$-strongly aiming | $\mu$-strongly convex |
|---|---|---|---|
| Must be $< 1$ | $\varepsilon\sqrt{1 + R^2}$ | $\varepsilon\sqrt{1 + 4R^2}$ | $\varepsilon\sqrt{1 + 5R^2 + 4R^4}$ |
| Constant | $(1 - \varepsilon\sqrt{1 + R^2})^2/(1 + \varepsilon)$ | $1 - \varepsilon\sqrt{1 + 4R^2}$ | $1 - \varepsilon\sqrt{1 + 5R^2 + 4R^4}$ |

Evidently, the geometric conditions and associated constants are quite different if $R \gg 1$. See Figure 1 for an illustration of $f$. Further details for the example and a comparison to less common notions such as quasar-convexity are given in Appendix C.2. We note that the example exploits the fact that $f$ is $C^{1,1}$- but not $C^2$-smooth: For $C^2$-functions, Rebjock & Boumal (2023) prove that the PL condition locally implies strong aiming condition.

The next example is trivial, but useful to illustrate why tangential movement should not matter.

*Example* 2. Let $\tilde{f}_1 : \mathbb{R}^{d-k} \to [0, \infty)$ be a non-negative $\mu$-strongly convex function such that $\tilde{f}_1(0) = 0$ and let $\tilde{f}_2 : \mathbb{R}^k \to [a, \infty)$ be a continuous function for $a > 0$. Define

$$f : \mathbb{R}^d \to \mathbb{R}, \qquad f(x) = \tilde{f}_2(x_1, \ldots, x_k) \cdot \tilde{f}_1(x_{k+1}, \ldots, x_d).$$

Then $f$ is $a\mu$-strongly aiming $\pi(x) = (x_1, \ldots, x_k, 0, \ldots, 0)$, but not strongly convex since the minimizer is non-unique. Similarly, if $A : \mathbb{R}^k \to \mathbb{R}^{(d-k)\times(d-k)}$ is a function which takes values in the set of symmetric matrices with eigenvalues larger than $\mu$, then

$$f : \mathbb{R}^d \to \mathbb{R}, \qquad f(x) = \frac{1}{2}(x_{k+1}, \ldots, x_d) A_{(x_1, \ldots, x_k)} \cdot (x_{k+1}, \ldots, x_d)^T$$

is $\mu$-strongly aiming, but generally non-convex.

*Example* 3. Let $\mathcal{M}$ be a compact $C^k$-submanifold of $\mathbb{R}^d$, $k \geq 2$ and $d(x) := \text{dist}(x, \mathcal{M})$. Then there exists a 'tubular neighborhood' $\mathcal{U}_\varepsilon = \{x \in \mathbb{R}^d : d(x) < \varepsilon\}$ on which $d$ is $C^k$-smooth and the unique closest point projection $\pi$ is well-defined and $C^{k-1}$-smooth – see Appendix A.

Assume that $f : \mathcal{U}_\varepsilon \to \mathbb{R}$ is given by $f(x) = \frac{\mu}{2} d(x)^2$ (and extended arbitrarily to $\mathbb{R}^d \setminus \mathcal{U}_\varepsilon$). Recall that $\nabla d(x)$ is the unit vector pointing towards the closest point in $\mathcal{M}$ at all points $x$ where the distance function is smooth, so in particular $\|\nabla d(x)\| = 1$. Thus $\pi(x) = x - d(x)\nabla d(x)$ and

$$\nabla f(x) \cdot (x - \pi(x)) = \mu\, d(x)\nabla d(x) \cdot \big(x - (x - d(x)\nabla d(x))\big) = \mu\, d^2(x)\, \|\nabla d(x)\|^2$$
$$= \frac{\mu}{2}\, d^2(x) + \frac{\mu}{2}\, d^2(x) = f(x) - f(\pi(x)) + \frac{\mu}{2}\, \|x - \pi(x)\|^2$$

for $x \in \mathcal{U}_\varepsilon$, i.e. $f$ is $\mu$-strongly aiming. On the other hand, $f$ is not convex unless $\mathcal{M}$ is. Otherwise, take $x_1, x_2 \in \mathcal{M}$ and $t \in (0, 1)$ such that $tx_1 + (1-t)x_2 \notin \mathcal{M}$. Then the map $t \mapsto d^2(tx_1 + (1-t)x_2)$ attains a maximum inside the interval $(0, 1)$, meaning that $d^2$ cannot be convex.

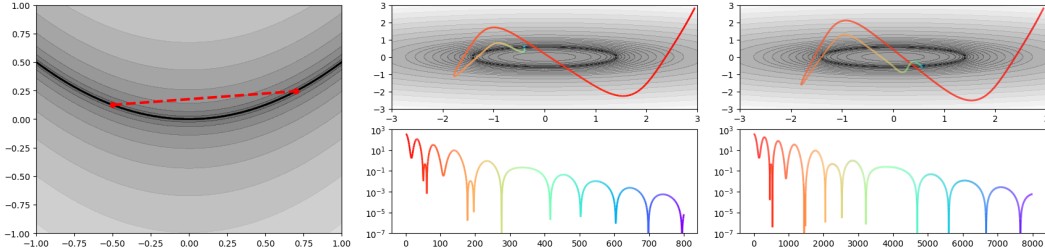

Figure 2: **Left:** The dashed red line connects two minimizers of the function $f$. Along the line, $f$ must achieve an interior local maximum. At this point, the Hessian $D^2 f$ cannot be positive definite. **Middle, Right:** Optimization trajectories for Nesterov's method (top) and its associated energy curve (bottom). The selection of limit point may depend crucially on optimization parameters: In the middle plot, we take 800 steps with stepsize $10^{-2}$ while on the right, we take 8,000 steps with stepsize $10^{-3}$ from the same initial point. The decay of $f(x_t)$ is similar for both trajectories, but the limit points on the manifold of minimizers are far apart. The objective function is $f(x, y) = (x^2/2 + 3y^2 - 1)^2$.

This consideration more generally shows that if the manifold of minimizers $\mathcal{M}$ of a function $f$ is not perfectly straight, then the objective function cannot be convex – see also Figure 2. More precisely:

**Lemma 4.** *(Wojtowytsch, 2023, based on Appendix B) Let $f : \mathbb{R}^d \to \mathbb{R}$ be a $C^2$-function and $\mathcal{M} = \{x \in \mathbb{R}^d : f(x) = \inf f\}$. Assume that $\mathcal{M}$ is a $k$-dimensional $C^1$-submanifold of $\mathbb{R}^d$, $z \in \mathcal{M}$, $T_z \mathcal{M}$ the tangent space at $z$, $r > 0$. If $\mathcal{M} \cap B_r(z)$ is not the same set as $(z + T_z \mathcal{M}) \cap B_r(z)$, then there exists $x \in B_r(z)$ such that $D^2 f(x)$ has a strictly negative eigenvalue.*

## 2.3 CONNECTION TO DEEP LEARNING

An important class of objective functions are those which combine the geometric features of Examples 2 and 3. Such functions can be seen as geometric prototypes for loss functions in overparametrized regression problems, such as in deep learning. Namely, consider a parametrized function class $h : \mathbb{R}^p \times \mathbb{R}^d \to \mathbb{R}$ of weights $w \in \mathbb{R}^p$ and data $x \in \mathbb{R}^d$ (e.g. a neural network) and the mean squared error (MSE) loss function

$$L_y : \mathbb{R}^p \to [0, \infty), \qquad L_y(w) = \frac{1}{2n} \sum_{i=1}^{n} \big(h(w, x_i) - y_i\big)^2, \qquad y = (y_1, \ldots, y_n) \in \mathbb{R}^n. \quad (2)$$

If $h$ is sufficiently smooth in $w$ and for every vector $y \in \mathbb{R}^n$, there exists $w_y \in \mathbb{R}^p$ such that $L_y(w_y) = 0$, then Cooper (2021) showed that for Lebesgue-almost all $y \in \mathbb{R}^n$, the set $\mathcal{M}_y = \{w \in \mathbb{R}^p : L_y(w) = 0\}$ is a $p - n$-dimensional submanifold of $\mathbb{R}^p$. Essentially, the solution set of $n$ equations $h(w, x_i) = y_i$ in $p$ variables is $p - n$-dimensional, much like when $h$ is linear in $w$.

Cooper (2021) demonstrates that the expressivity and smoothness assumptions provably apply to parametrized function classes $h(w, x)$ of sufficiently wide neural networks with analytic activation function such as tanh or sigmoid. Cooper (2021)'s proof involves the regular value theorem and Sard's theorem to show that all gradients $\nabla h_w(w, x_i)$ are linearly independent on almost every level set $\mathcal{M}_y$. As a byproduct, this implies that the Hessian of the loss function

$$D^2 L(w) = \frac{1}{n} \sum_{i=1}^{n} \bigg( \underbrace{\big(h(w, x_i) - y_i\big)}_{=0} D_w^2 h(w, x_i) + \nabla_w h(w, x_i) \otimes \nabla_w h(w, x_i) \bigg)$$

has full rank $n$ at every $x \in \mathcal{M}_y$ (for almost every $y$). Thus, all $n$ eigenvalues in direction orthogonal to $\mathcal{M}$ are non-zero. We prove the following in Appendix A. The same connection has been made e.g. by Rebjock & Boumal (2023) for $C^2$-functions and overparametrized regression problems.

**Lemma 5.** *Assume that $f : \mathbb{R}^d \to \mathbb{R}$ is $C^2$-smooth and that $\mathcal{M} = \{x \in \mathbb{R}^d : f(x) = \inf f\}$ is a closed $k$-dimensional $C^2$-submanifold of $\mathbb{R}^d$ (i.e. compact and without boundary). If $D^2 f(x)$ has rank $d - k$ everywhere on $\mathcal{M}$, then there exist $\mu, \alpha > 0$ such that there exists a $C^1$-smooth closest point projection $\pi : U_\alpha \to \mathbb{R}$ with $U_\alpha = \{x : f(x) < \alpha\}$ and $f$ is $\mu$-strongly aiming with $\pi$.*

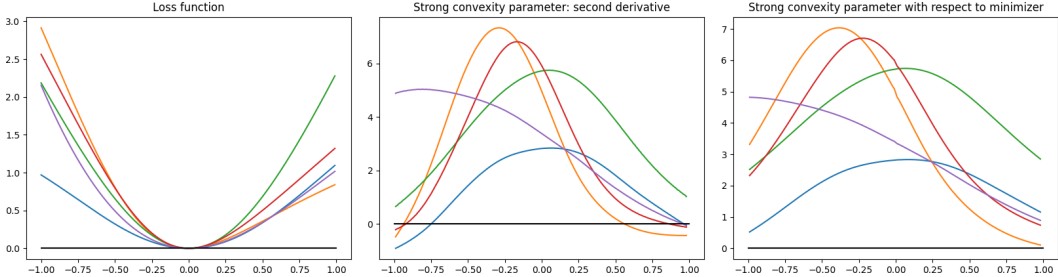

Figure 3: Convexity analysis of $\phi(t) = L(w + tg)$ for $w$ near global minimizers of a loss function $L$ and $g = \nabla L(w)/\|\nabla L(w)\|$. Left: $\phi(t)$, middle: second derivative of $\phi$, right: estimated strong aiming parameter $\mu$ for $t \in [-1, 1]$. Evidently, $\phi$ is strongly convex in a neighborhood of the minimizers. Strong aiming condition yields consistently larger constants than the strong convexity parameter obtained from second derivatives. Different colors correspond to different random initializations.

In particular, $f, \mathcal{M}$ meet all conditions in Section 2.1. Note that by Lemma 4, the loss function $f$ cannot be convex since a compact manifold cannot be perfectly straight everywhere. Lemma 5 does not apply to networks with the non-smooth ReLU activation $\sigma(z) = \max\{z, 0\}$, also here minimizers cannot be isolated due to the continuous scaling symmetry $\sigma(z) = \lambda^{-1}\sigma(\lambda z)$ for $\lambda > 0$.

Wojtowytsch (2023, Theorem 2.6) shows that the assumption that $\mathcal{M}$ is compact is a simplification and generally does not apply in deep learning. Local versions of Lemma 5 could be proved with $\mu, \alpha$ which are positive functions on the manifold, but not necessarily bounded away from zero. Naturally, this suffices in all cases where we provably remain in a local neighborhood in the course of optimization. We eschew this greater generality for the sake of geometric clarity and commit the pervasive sin of optimization theory for deep learning: We make global assumptions which can only be guaranteed locally. For a further comparison of geometric conditions in optimization and deep learning, see also Appendix C.

We illustrate in Figure 3 that our assumptions are locally reasonable in deep learning. We trained a fully connected neural network (with 10 layers, width 35, tanh activation) to fit labels $y_i$ at 100 randomly generated datapoints $x_i \in \mathbb{R}^{12}$. The small dataset size allowed us to use the exact gradient and loss function instead of stochastic approximations, for a better exploration of the loss landscape. Since the closest minimizer is generally unknown, we use the gradient as a proxy and examine the convexity of $\phi(t) = L(w + tg)$ for $w$ very close to the set of global minimizers of the loss function $L$ as in (2) and $g = \nabla L(w)/\|\nabla L(w)\|$. Labels were generated using a randomly initialized teacher network (with 7 layers and width 20). Student networks were trained for 10,000 epochs using stochastic gradient descent with Nesterov momentum, with learning rate $\eta = 0.005$ and momentum $\rho = 0.99$. Final training loss ranged between $10^{-12}$ and $10^{-9}$ across the five runs. Second derivatives were approximated using second order difference quotients $\phi''(t) \approx \frac{\phi(t+h)-2\phi(t)+\phi(t-h)}{h^2}$ for $h = 0.01$. Similarly, the strong aiming parameter with respect to the global minimizer was estimated by $2\frac{\phi'(t)t-\phi(t)+\inf \phi}{t^2}$ where $\phi'(t)$ was estimated as $\frac{\phi(t+h)-\phi(t-h)}{2h}$.

Due to the inherent randomness of training, geometric behaviors varied. Generally, the loss function was convex along the gradient direction near minimizers, but neighborhood size, magnitude of the second derivative, and steepness of the loss function varied. Some runs exhibited convex but not strongly convex behavior, while others failed to reach zero loss along the line $w + tg/\|g\|$. The gradient is an imperfect approximation of the minimizer's direction, and rounding errors may occur when it is so small near the set of minimizers.

## 3 MAIN CONTRIBUTIONS

### 3.1 OPTIMIZATION IN CONTINUOUS TIME

In this section, we study a continuous time version of gradient descent with (heavy ball or Nesterov) momentum derived by Su et al. (2016). Namely, we study solutions of the heavy ball ODE

$$\ddot{x} + \gamma \dot{x} = -\nabla f(x), \qquad x(0) = x_0, \qquad \dot{x}(0) = 0. \tag{3}$$

This is a popular model for the study of accelerated methods in optimization which avoids some of the technicalities of discrete time stepping algorithms while relying on the same core geometric concepts. Our main result is the following.

**Theorem 6.** *[Continuous time convergence guarantee] Assume that $f$ satisfies the assumptions of Section 2.1, $\gamma = 2\sqrt{\mu}$ and that $x_0$ satisfies $f(x_0) < \alpha$ where $\alpha$ is as in Section 2.1. Then there exists a unique solution $x(t) = x_t$ of (3) and $f(x_t) < \alpha$ for all $t > 0$. Furthermore, $x$ is $C^2$-smooth and*

$$f(x_t) - \inf_{z \in \mathbb{R}^d} f(z) \le e^{-\sqrt{\mu}\, t} \left( f(x_0) - \inf f + \frac{\mu}{2} \operatorname{dist}(x_0, \mathcal{M})^2 \right).$$

The proof can be found in Appendix B. There, we prove that the Lyapunov function

$$\mathcal{L}(t) = f(x_t) - f(\pi(x_t)) + \frac{1}{2} \left\| \dot{x}_t + \sqrt{\mu}\big(x_t - \pi(x_t)\big) \right\|^2$$

satisfies $\mathcal{L}(t) \le e^{-\sqrt{\mu} t}\mathcal{L}(0)$. The function $\mathcal{L}$ can be considered as a modified energy with potential energy $f - \inf f + \frac{\mu}{2}\|x - \pi(x)\|^2$ and kinetic energy $\frac{1}{2}\|\dot{x}\|^2$, but with a more complex quadratic term in which velocity and position interact. Going back to (Su et al., 2016), this strategy of proof is common in convex optimization. The main difficulty in our more general geometric setting is that the time derivative of $\pi(x_t)$ (i.e. tangential velocity) is not zero in general, unless the minimizer is unique in $\mathcal{U}_\alpha$. Thus, we have to additionally control for the interaction of the tangential velocity with the other terms in the Lyapunov sequence. One key observation that helps us is that the connecting line $x_t - \pi(x_t)$ is orthogonal to the velocity of $\pi(x_t)$. The other technical tool is the following lemma.

**Lemma 7.** *Let $\mathcal{M}$ be a $C^2$-submanifold of $\mathbb{R}^d$ and $\mathcal{U}$ an open set containing $\mathcal{M}$ such that there exists a unique closest point projection $\pi : \mathcal{U} \to \mathcal{M}$. Let $x : (-\varepsilon, \varepsilon) \to \mathcal{U}$ be a $C^1$-curve and $z(t) := \pi \circ x(t)$. Then $\langle \dot{x}, \dot{z} \rangle \ge 0$ on $(-\varepsilon, \varepsilon)$.*

Geometrically, Lemma 7 states that the closest point projection $z$ of a point $x$ does not move in the opposite direction when we move $x$. Despite its geometric simplicity, Lemma 7 is non-trivial and a crucial ingredient in our proofs. Its proof is given in Appendix A.

Venturi et al. (2019) show that loss functions in overparametrized deep learning do not have strict local minimizers outside the set of global minimizers (under suitable assumptions). Under a quantitative version of this geometric assumption, we obtain a global version of Theorem 6, albeit with less precise quantitative guarantees.

**Theorem 8.** *In addition to the assumptions of Section 2.1, assume that*

1. *for every $R > \inf f$, there exists $L_R > 0$ such that $\nabla f$ is Lipschitz-continuous with Lipschitz constant $L_R$ on $\mathcal{U}_R = \{x : f(x) < R\}$ and*

2. *there exists a value $\delta > 0$ such that $\|\nabla f(x)\|^2 < \delta$ implies that $f(x) - \inf f < \frac{\alpha}{2}$.*

*Then, for any $x_0 \in \mathbb{R}^d$, there exists $T \ge 0$ such that $x(t) \in \mathcal{U}_\alpha$ for all $t \ge T$ and such that $f(x(t)) - \inf f \le \big(3\alpha/2 + \operatorname{dist}(x_T, \mathcal{M})^2\big)e^{\sqrt{\mu}(T-t)}$ for all $t > T$.*

The rate of decay here is $e^{-\sqrt{\mu} t}$, as compared to $e^{-\mu t}$ for gradient flow. We see more clearly why we speak of acceleration in discrete time (see the discussion below Theorem 11).

*Remark 9.* We conjecture that Theorem 6 remains valid if the set $\mathcal{M}$ is convex rather than a smooth manifold. Then $\pi$ is defined globally, but generally only Lipschitz-continuous and not smooth.

## 3.2 DETERMINISTIC OPTIMIZATION IN DISCRETE TIME

We can equivalently write the heavy ball ODE (3) as a system of two first order ODEs:

$$\begin{cases} \dot{x} &= v \\ \dot{v} &= -2\sqrt{\mu}\, v - \nabla f(x), \end{cases} \qquad x(0) = x_0, \qquad v(0) = 0. \tag{4}$$

We choose a time-stepping scheme

$$x'_n = x_n + \sqrt{\eta} v_n, \qquad g_n = \nabla f(x'_n), \qquad x_{n+1} = x'_n - \eta g_n, \qquad v_{n+1} = \rho\big(v_n - \sqrt{\eta} g_n\big) \tag{5}$$

for $\rho = \frac{1-\sqrt{\mu\eta}}{1+\sqrt{\mu\eta}} = 1 - 2\sqrt{\mu\eta} + O(\eta)$. In particular, we have

$$x_{n+1} = x_n + \sqrt{\eta}\, v_n + O(\eta), \qquad v_{n+1} = v_n - \sqrt{\eta}\big(2\sqrt{\mu}\, v_n + \nabla f(x_n)\big) + O(\eta),$$

i.e. the scheme is a time discretization of the heavy ball system (4) with time-step size $\sqrt{\eta}$. The square root is chosen for consistency with the literature. This scheme is a geometrically intuitive reparametrization of Nesterov's accelerated gradient descent algorithm, except for the fact that Nesterov's scheme typically begins with the gradient descent step rather than the momentum step (see Gupta et al., 2024, Appendix B, for a proof of equivalence).

In discrete time, we need to make additional quantitative regularity assumptions on both $f$ and the projection map $\pi$ in order to ensure that the time step size is sufficiently small to recover the continuous time behavior. Note that the Hessian of the objective function $f$ is positive semi-definite on the set of global minimizers $\mathcal{M}$, i.e. it stands to reason that any negative eigenvalues of $D^2 f$ should be small, at least close to $\mathcal{M}$. We note the following.

**Lemma 10.** *Assume that $D^2 f(x) \geq -\varepsilon$ in a ball $B_r(x_0)$. Then*

$$\langle \nabla f(x), x - z \rangle \geq f(x) - f(z) - \frac{\varepsilon}{2}\|x - z\|^2 \qquad \forall\, x, z \in B_r(x_0).$$

This lemma informs our geometric assumptions on the objective function $f$. For the closest point projection $\pi$, we assume that $\pi(x) = Px + x^*$ for some (linear) orthogonal projection $P$ onto a subspace $V \subseteq \mathbb{R}^d$ and a fixed vector $x^*$ which is the unique element with the smallest Euclidean norm in the affine space $\tilde{V} = x^* + V$. The assumption that the derivative $D\pi \equiv P$ is constant is a (very restrictive) geometric linearization, and relaxing it is an important subject of future research. Still, it applies to many functions which are not convex, such as those with a unique minimizer (i.e. $P \equiv 0$) or those in Example 2. With an eye towards stochastic optimization, we opt for the simpler global geometric assumptions, see also Remark 16.

**Theorem 11.** *Assume that $f$ is $L$-smooth and the sequences $x_n, x'_n, v_n$ are generated according to the Nesterov scheme (5) with parameters $\eta \leq 1/L$ and $\rho = (1 - \sqrt{\mu\eta})/(1 + \sqrt{\mu\eta})$. Assume further that there exists an affine linear projection map $\pi(x) = Px + x^*$ such that*

$$\langle \nabla f(x), x - \pi(x) \rangle \geq f(x) - f(\pi(x)) + \frac{\mu}{2}\|x - \pi(x)\|^2. \tag{6}$$

*Finally, assume that for arbitrary $x, v \in \mathbb{R}^d$ we have*

$$\langle \nabla f(x+v), v \rangle \geq f(x+v) - f(x) - \frac{\varepsilon}{2}\|v\|^2 \tag{7}$$

*with some $\varepsilon \leq \sqrt{\mu/\eta}$. Then*

$$f(x_n) - \inf f \leq (1 - \sqrt{\mu\eta})^n \left[ f(x_0) - \inf f + \frac{\mu}{2}\|x_0 - \pi(x_0)\|^2 \right].$$

The proof, given in Appendix B.2, builds on similar ideas as Theorem 6, with additional complications introduced by the discrete-time setting. We have to modify the usual Lyapunov sequence used for strongly convex functions. We treat the tangential and normal components of the velocity (i.e. $Pv_n$ and $P^\perp v_n = (I - P)v_n$) separately, and carefully choose coefficients to ensure that the following Lyapunov sequence decays at each step,

$$\mathcal{L}_n = f(x_n) - \inf f + \frac{1}{2}\| P^\perp v_n + \sqrt{\mu}(x'_n - \pi(x'_n))\|^2 + \frac{(1 + \sqrt{\mu\eta})^2}{2(1 - \sqrt{\mu\eta})}\|Pv_n\|^2.$$

In Theorem 11, we see more clearly than in Theorem 6 why we talk of acceleration: While gradient descent would achieve a decay rate of $(1 - \mu/L)^n$ with the commonly proposed step size $\eta = 1/L$ in discrete time based on our assumptions, Nesterov's method achieves decay like $(1 - \sqrt{\mu/L})^n$ with $\eta = 1/L$. Since $\mu/L \leq 1$, Nesterov's method converges much faster than gradient descent.

*Remark 12.* Note that the negative eigenvalues of the Hessian may be as large as $\sqrt{\mu/\eta}$ in Theorem 11. If $\eta$ is chosen as large as $1/L$, this is a real restriction of the eigenvalues to the range $[-\sqrt{\mu L}, L]$. However, since the Hessian eigenvalues of an $L$-smooth function are in $[-L, L]$ a priori, there is no additional restriction if $\sqrt{\mu/\eta} \geq L$, i.e. if $\eta < \frac{\mu}{L^2}$. This corresponds to the continuous time guarantee of Theorem 6, which does not depend on the magnitude of the negative eigenvalues.

However, if the eigenvalues of $D^2 f$ are as negative as $-L$, the step size $\eta = \mu/L^2$ is so small that it does not improve upon gradient descent with step size $\eta = 1/L$ since $\sqrt{\mu\eta} = \mu/L$ in this case. Thus, if $f$ is too far from being convex, acceleration may not be achievable in discrete time. Notably, especially close to the set of global minimizers, we can picture $\varepsilon$ as small compared to $L$.

### 3.3 STOCHASTIC OPTIMIZATION IN DISCRETE TIME

In typical applications in deep learning, the gradient $\nabla f$ of the objective function/loss function $f$ is prohibitively expensive to evaluate, but we have access to stochastic estimates of the true gradient. In this section, in addition to the assumptions of Theorem 11, we assume that we are given a probability space $(\Omega, \mathcal{A}, \mathbb{Q})$ and a measurable function $g : \mathbb{R}^d \times \Omega \to \mathbb{R}^d$ such that

$$\mathbb{E}_{\omega \sim \mathbb{Q}}\big[g(x,\omega)\big] = \nabla f(x), \qquad \mathbb{E}_{\omega \sim \mathbb{Q}}\big[\|g(x,\omega)\|^2\big] < +\infty \qquad \forall\, x \in \mathbb{R}^d. \tag{8}$$

For quantitative statements, a more precise assumption on the variance of the gradient estimates must be made. We make the modelling assumption

$$\mathbb{E}_{\omega \sim \mathbb{Q}}\big[\|g(x,\omega) - \nabla f(x)\|^2\big] \leq \sigma_a^2 + \sigma_m^2\|\nabla f(x)\|^2 \qquad \forall\, x \in \mathbb{R}^d. \tag{9}$$

We call $\sigma_a$ the additive standard deviation and $\sigma_m$ the multiplicative standard deviation since they resemble the prototypical example $g = (1 + \sigma_m N_1)\nabla f + \sigma_a N_2$ where $N_1, N_2$ are random variables with mean zero and variance one. The case of purely additive noise (i.e. $\sigma_m = 0$) is classical and hails back to the seminal article of Robbins & Monro (1951). The case of purely multiplicative noise is much closer to reality in overparametrized learning: If all data points can be fit exactly, there is no noise when estimating the gradient of the empirical risk/training loss on the set of global minimizers. It has received significant attention more recently by Liu & Belkin (2018); Bassily et al. (2018); Vaswani et al. (2019); Even et al. (2021); Wojtowytsch (2023); Gupta et al. (2024) and others.

Let us consider the purely additive case first. We follow the scheme (5), but we replace the deterministic gradient $\nabla f(x'_n)$ by $g_n = g(x'_n, \omega_n)$ where $\omega_0, \omega_1, \dots$ are drawn from $\Omega$ independently of each other and the initial condition $x_0$ with law $\mathbb{Q}$. This framework allows e.g. for minibatch sampling (but assumes that all batches are drawn independently of each other from the dataset).

**Theorem 13.** *[Acceleration with additive noise] Assume that $f, P$ are as in Theorem 11 and that the $g$ satisfies (8) and (9) with $\sigma_m = 0$. Assume that the sequences $x_n, x'_n, v_n$ are generated by the scheme (5) for parameters $\eta \leq 1/L$ and $\rho = \frac{1 - \sqrt{\mu\eta}}{1 + \sqrt{\mu\eta}}$, but with the stochastic gradient estimates $g(x'_n, \omega_n)$ with independently identically distributed $\omega_n$ in place of $\nabla f(x'_n)$. Then*

$$\mathbb{E}[f(x_n) - \inf f] \leq (1 - \sqrt{\mu\eta})^n \left[ f(x_0) - \inf f + \frac{\mu}{2}\|x_0 - \pi(x_0)\|^2 \right] + \frac{\sigma_a^2 \sqrt{\eta}}{\sqrt{\mu}}.$$

Thus Nesterov's method reduces $f(x_n)$ below a 'noise level' proportional to $\sigma_a^2 \sqrt{\eta/\mu}$ at a linear rate $(1 - \sqrt{\mu\eta})^n$ in the presence of additive noise. The analogous bound for stochastic gradient descent was obtained (in the more general setting of PL functions) in (Karimi et al., 2016, Theorem 4) as

$$\mathbb{E}[f(x_n) - \inf f] \leq (1 - \mu\eta)^k \,\mathbb{E}\big[f(x_0) - \inf f\big] + \frac{L\sigma_a^2 \eta}{2\mu}.$$

With the largest admissible learning rate $\eta = 1/L$, the noise level for GD is $\sigma_a^2/2\mu$ compared to the usually much lower value $\sigma_a^2/\sqrt{L\mu}$ for Nesterov's method. Keeping a memory of previous gradient estimates facilitates 'averaging out' the random noise.

With a fixed positive learning rate $\eta$, generally $f(x_n) \not\to 0$ unless $\sigma_a = 0$. We therefore consider a sequence of decreasing step sizes.

**Theorem 14.** *[Additive noise and decreasing step size] Assume that $f, g$ are as in Theorem 13 and that the sequences $x_n, x'_n, \rho_n$ are generated by the scheme*

$$x'_n = x_n + \sqrt{\eta_{n-1}}\, v_n, \qquad x_{n+1} = x'_n - \eta_n g_n, \qquad v_{n+1} = \rho_n(v_n - \sqrt{\eta_n}\, g_n)$$

*for parameters $\eta_n = \frac{\mu}{(n + \sqrt{L\mu} + 1)^2}$, $\rho_n = \frac{1 - \sqrt{\mu\eta_n}}{1 + \sqrt{\mu\eta_n}}$. If $\varepsilon \leq \sqrt{\mu/\eta_0} = \mu + \sqrt{L\mu}$, then*

$$\mathbb{E}\big[f(x_n) - \inf f\big] \leq \frac{\sqrt{\frac{L}{\mu}}\, \mathbb{E}\big[f(x_0) - \inf f + \frac{1}{2}\|x_0 - \pi(x_0)\|^2\big] + \frac{\sigma_a^2}{\mu} \log\big(1 + n\sqrt{\mu/L}\big)}{n + \sqrt{L/\mu}}.$$

Note that the 'physical' step size $\sqrt{\eta_n}$ decays as $1/n$ and thus satisfies the (non-)summability conditions of Robbins & Monro (1951). We can allow for larger $\varepsilon$ by choosing $\eta_n = \mu/(n + n_0)$ with $n_0 > \sqrt{L/\mu} - 1$ for a smaller initial step size.

If $\sigma_m > 0$, Liu & Belkin (2018) and Gupta et al. (2024) show that Nesterov's scheme no longer achieves acceleration. For general noise, we therefore consider a modified Nesterov scheme:

$$x'_n = x_n + \sqrt{\alpha}v_n, \qquad x_{n+1} = x'_n - \eta g_n, \qquad v_{n+1} = \rho_n\big(v_n - \sqrt{\alpha}g_n\big) \tag{10}$$

where again $g_n = g(x'_n, \omega_n)$. The scheme (10) was introduced as the Accelerated Gradient method with Noisy EStimators method (AGNES) by Gupta et al. (2024) in convex and strongly convex optimization with purely multiplicative noise. Compared to Nesterov's algorithm, AGNES has an additional parameter $\alpha$ which is required to adapt to the multiplicative variance, at least if $\sigma_m \geq 1$. Here, we generalize the work of Gupta et al. (2024) by both allowing noise with general scaling for $\sigma_a, \sigma_m > 0$ and relaxing the convexity assumption on $f$.

**Theorem 15.** *[Additive and multiplicative noise] Assume that $f, P, x^*$ are as in Theorem 11 and that $g$ is a family of gradient estimators such that (8) and (9) hold for some $\sigma_a, \sigma_m \geq 0$. Assume that the sequences $x_n, x'_n, v_n$ are generated by the AGNES scheme (10) with parameters*

$$0 < \eta \leq \frac{1}{L(1 + \sigma_m^2)}, \qquad \rho = \frac{1 - \sqrt{\frac{\mu\eta}{1 + \sigma_m^2}}}{1 + \sqrt{\frac{\mu\eta}{1 + \sigma_m^2}}}, \qquad \alpha = \frac{1 - \sqrt{\mu(1 + \sigma_m^2)\eta}}{1 - \sqrt{\mu(1 + \sigma_m^2)\eta} + \sigma_m^2}\,\eta.$$

*Then, if $\varepsilon < \sqrt{\mu(1 + \sigma_m^2)/\eta}$, we have*

$$\mathbb{E}\big[f(x_n) - \inf f\big] \leq \left(1 - \sqrt{\frac{\mu\eta}{1 + \sigma_m^2}}\right)^n \mathbb{E}\left[f(x_0) - \inf f + \frac{\mu}{2}\|x_0 - \pi(x_0)\|^2\right] + \frac{\sigma_a^2\sqrt{\eta}}{\sqrt{\mu(1 + \sigma_m^2)}}.$$

The proof of Theorem 15 is given in Appendix B.4. While at a glance it appears that the multiplicative noise is helping us reduce the additive error term, this is merely a consequence of the small learning rate which is forced upon us. The condition on the negative eigenvalues is relaxed to $\varepsilon \leq \sqrt{\mu L}(1 + \sigma_m^2)$ with the largest admissible step size $\eta = 1/(L(1 + \sigma_m^2))$ as the issues stemming from tangential drift pale in comparison to those stemming from stochastic gradient estimates. For comparison, if we chose the same $\eta$ in Theorem 11, we could only allow for $\varepsilon \leq \sqrt{\mu L}\sqrt{1 + \sigma_m^2} < \sqrt{\mu L}(1 + \sigma_m^2)$, but we would obtain a rate of convergence of $1 - \sqrt{\mu/L(1 + \sigma_m^2)}$. In the stochastic case, we only achieve $1 - \sqrt{\mu/L}/(1 + \sigma_m^2)$. Thus the limiting factor is the stochastic noise, not the geometry of $f$.

*Remark* 16. We opted for a *linear* closest point projection map to facilitate proofs. In the non-linear case, closest point projections cannot be defined globally: Jessen (1940); Busemann (1947); Phelps (1957) show that if $K$ is a subset of $\mathbb{R}^d$ such that for every $x \in \mathbb{R}^d$ there exists a unique closest point in $K$, then $K$ is closed and convex. If $K$ is both a $k$-dimensional submanifold of $\mathbb{R}^d$ and a convex set, then $K$ is an affine $k$-dimensional subspace of $\mathbb{R}^d$, i.e. our assumptions are the most general when assuming that a unique closest point projection onto a submanifold is defined globally.

Thus, if $\mathcal{M}$ is not an affine space, we can only assume that $\pi$ is 'good' in a neighborhood of $\mathcal{M}$. In stochastic optimization, where we can randomly 'jump' out of the good neighborhood, this leads to serious technical challenges. Guarantees on 'remaining local' with high probability have recently been derived for SGD with additive noise and decaying learning rates by Mertikopoulos et al. (2020) and with multiplicative noise by Wojtowytsch (2023). To avoid obscuring the new geometric constructions, we opted to forgo this highly technical setting here and prioritize the extension of Theorems 6 and 11 towards stochastic optimization.

## 4 CONCLUSION

We have proved that first order momentum-based methods accelerate convergence in a more general setting than convex optimization with many geometric features motivated by loss landscapes encountered in deep learning. The models we studied include the heavy ball ODE and deterministic and stochastic optimization schemes in discrete time under various noise assumptions.

The most cumbersome limitation of our convergence guarantees is the assumption that the closest point projection onto the set of minimizers is affine linear in the discrete time setting, i.e. the derivative $D\pi$ is constant in space. Future work will focus on relaxing this assumption.

ACKNOWLEDGEMENTS

The authors gratefully acknowledge the financial support of the NSF, grant DMS 2424801. The authors are grateful to Quentin Merigot, who pointed out a simpler proof of Lemma 7 which requires less regularity than the authors' original approach.

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

## A  PROOFS OF LEMMAS 5 AND 7: GEOMETRY OF THE ENERGY LANDSCAPE

We assume that the reader is familiar with basic concepts of differential geometry, such as submanifolds of Euclidean spaces and tangent spaces and with concepts of multi-variate analysis such as compactness and the inverse function theorem.

We first recall an important observation: Assume that $\mathcal{M}$ is a $C^1$-manifold. Fix a point $x \in \mathbb{R}^d$. Assume that the function $d : \mathcal{M} \to \mathbb{R}$ given by $d(z) = \|x - z\|$ has a local extremum at $z \in \mathcal{M}$. Then for every $C^1$-curve $\gamma : (-\varepsilon, \varepsilon) \to \mathcal{M}$ such that $\gamma(0) = z$, we have

$$0 = \frac{d}{dt}\Big|_{t=0} d(\gamma(t))^2 = 2\langle x - \gamma(0), \dot{\gamma}(0)\rangle = 2\langle x - z, \dot{\gamma}(0)\rangle$$

or in other words: the connecting line $x - z$ is orthogonal to the tangent space $T_z\mathcal{M}$. This is in particular true if $z$ is the closest point in $\mathcal{M}$ to $x$.

**Lemma 7.** *Let $\mathcal{M}$ be a $C^2$-submanifold of $\mathbb{R}^d$ and $\mathcal{U}$ an open set containing $\mathcal{M}$ such that there exists a unique closest point projection $\pi : \mathcal{U} \to \mathcal{M}$. Let $x : (-\varepsilon, \varepsilon) \to \mathcal{U}$ be a $C^1$-curve and $z(t) := \pi \circ x(t)$. Then $\langle \dot{x}, \dot{z}\rangle \geq 0$ on $(-\varepsilon, \varepsilon)$.*

*Proof.* The closest point projection onto a $C^2$-manifold is $C^1$-smooth and satisfies

$$\pi(x) = x - d(x)\,\nabla d(x) \tag{11}$$

where $d(x) = \operatorname{dist}(x, \mathcal{M})$ is the distance function to the manifold, i.e. $\nabla d(x)$ gives the unit vector pointing directly towards the manifold. For details, see the proof of Lemma 5.

We can rewrite (11) as

$$\pi(x) = \nabla\left(\frac{\|x\|^2}{2} - \frac{d(x)^2}{2}\right) = \nabla\left(\frac{\|x\|^2}{2} - \frac{\min_{x\in\mathcal{M}}\|x - z\|^2}{2}\right)$$

$$= \nabla_x \max_{\mathcal{M}}\left(\frac{\|x\|^2}{2} - \frac{\|x\|^2 - 2\langle x, z\rangle + \|z\|^2}{2}\right) = \nabla_x \max_{z\in\mathcal{M}}\left(\langle x, z\rangle - \frac{\|z\|^2}{2}\right).$$

We are taking the (pointwise in $x$) maximum over a class of functions which are linear in $x$, i.e. we find that

$$\xi(x) := \max_{z\in\mathcal{M}}\left(\langle x, z\rangle - \frac{\|z\|^2}{2}\right)$$

is convex. In particular, the derivative matrix $D\pi = D^2\xi$ is symmetric and positive semi-definite and thus

$$\left\langle \dot{x}, \frac{d}{dt}\pi \circ x\right\rangle = \langle \dot{x}, \, D\pi(x)\,\dot{x}\rangle \geq 0. \qquad \square$$

**Lemma 5.** *Assume that $f : \mathbb{R}^d \to \mathbb{R}$ is $C^2$-smooth and that $\mathcal{M} = \{x \in \mathbb{R}^d : f(x) = \inf f\}$ is a closed $k$-dimensional $C^2$-submanifold of $\mathbb{R}^d$ (i.e. compact and without boundary). If $D^2 f(x)$ has rank $d - k$ everywhere on $\mathcal{M}$, then there exist $\mu, \alpha > 0$ such that there exists a $C^1$-smooth closest point projection $\pi : U_\alpha \to \mathbb{R}$ with $U_\alpha = \{x : f(x) < \alpha\}$ and $f$ is $\mu$-strongly aiming with $\pi$.*

*Proof.* This result uses standard ideas from differential geometry. For the reader's convenience, we provide a full proof sketch.

**Step 1. Closest point projection.** Assume that $\mathcal{M}$ is a $C^m$-manifold for $m \geq 2$. Fix a point $z_0 \in \mathcal{M}$, a radius $r > 0$ and the neighbourhood $U = B_r(z_0)$. We assume that $r > 0$ is so small that there exists a $C^m$-diffeomorphism $\phi : U \to V \subseteq \mathbb{R}^d$ such that $\phi(U \cap \mathcal{M}) = V \cap \{y : y_{k+1} = \cdots = y_d = 0\}$. If $\mathcal{M}$ is a $C^m$-manifold, we can find a collection of $C^{m-1}$-smooth vector fields $A_1, \ldots, A_d$ such that

1. $A_1(z), \ldots, A_d(z)$ span the tangent space $T_z\mathcal{M}$ for all $z \in \mathcal{M} \cap U$ and

2. $\langle A_i(z), A_j(z)\rangle = \delta_{ij}$ for all $i, j = 1, \ldots, d$ and $z \in \mathcal{M} \cap U$.

Such vector fields can be obtained for instance by applying the Gram-Schmidt algorithm to the columns of the derivative matrix $D(\phi^{-1})_{\phi(z)} = (D\phi_z)^{-1}$ of the inverse diffeomorphism. The algorithm returns an orthonormal basis since $\phi$ is a diffeomorphism, i.e. $D\phi$ has full rank. The first $k$ columns span the tangent space of $T_x\mathcal{M}$ since motion tangential to $\mathcal{M}$ in $U$ corresponds to motion where the last $d - k$ coordinates are kept zero in $V$, i.e. to the first $k$ columns of $D\phi^{-1}$.

We now introduce new coordinates: Denote by $\hat{V} \subseteq \mathbb{R}^k$ the set such that $\hat{V} \times \{0_{d-k}\} = V \cap \{y : y_{k+1} = \cdots = y_d = 0\}$ and

$$\Psi : \hat{V} \times \mathbb{R}^{d-k} \to \mathbb{R}^d, \qquad \Psi(\hat{y}, s_{k+1}, \ldots, s_d) = \phi^{-1}(\hat{y}, 0) + \sum_{i=k+1}^{d} s_i\, A_i\big(\phi^{-1}(\hat{y}, 0)\big).$$

The map $\Psi$ is $C^{m-1}$-smooth since it is linear in $s$ and the least regular components, the vector fields $A_i$, are $C^{m-1}$-smooth in $y$. If $m \geq 2$, we trivially find that $\Psi$ is differentiable and $D\Psi_{(\hat{y},0)} = (\partial_{\hat{y}_1}\phi, \ldots, \partial_{\hat{y}_k}\phi, A_{k+1}, \ldots, A_d)$ is invertible. Hence, the map $\Psi$ is a local diffeomorphism by the inverse function theorem. Notably, we see that

$$x - \Psi(\hat{y}, 0) \perp T_{\Psi(\hat{y},0)}\mathcal{M} \quad \Leftrightarrow \quad x - \Psi(\hat{y}, 0) \in \mathrm{span}\,\{A_{k+1}(\Psi(\hat{y}, 0)), \ldots, A_d(\Psi(\hat{y}, 0))\}$$

and thus if and only if

$$x = \Psi(\hat{y}, 0) + \sum_{i=k+1}^{d} s_i A_i\big(\Psi(\hat{y}, 0)\big) = \Psi(\hat{y}, s)$$

for some $s \in \mathbb{R}^{d-k}$. In a neighbourhood of $z_0$ where $\Psi$ is a diffeomorphism, we set $\pi(\Psi(\hat{y}, s)) = \Psi(\hat{y}, 0)$, i.e.

$$\pi = \Psi \circ P_{\mathbb{R}^k} \circ \Psi^{-1}, \qquad P(y_1, \ldots, y_d) = (y_1, \ldots, y_k, 0, \ldots, 0).$$

The map is as smooth as $\Psi$, i.e. $C^{m-1}$-smooth (assuming that $m \geq 2$). The map $\pi$ defined in this way may not be the unique closest point projection on all of $U$ (e.g. when a point $z' \in \mathcal{M} \setminus U$ is closer), but it is guaranteed to be the unique closest point projection on a smaller subset $B_{r/2}(z^*)$ where the closest point on $\mathcal{M}$ is closer than the boundary of $U$.

Thus, for every point $z \in \mathcal{M}$, there exists a neighborhood $B_{r(z)}(z)$ for $r(z) > 0$ in which a unique closest point projection is defined. Setting $U := \bigcup_{z\in\mathcal{M}} B_{r(z)}(z)$, we find a neighborhood of the manifold $\mathcal{M}$ inside of which the closest point projection is defined.

Assume that the radius $r(z)$ is chosen as the supremum of all admissible radii. Then the function $r(z)$ is strictly positive and Lipschitz-continuous with Lipschitz-constant 1: $r(z') \geq r(z) - \|z - z'\|$ since $B_{r-\|z-z'\|}(z') \subseteq B_r(z)$. Exchanging the role of $z, z'$ shows that

$$r(z') \geq r(z) - \|z - z'\|, \quad r(z) \geq r(z') - \|z - z'\| \quad \Rightarrow \quad |r(z) - r(z')| \leq \|z - z'\| \quad \forall\, z, z' \in \mathcal{M}.$$

In particular, if $\mathcal{M}$ is compact, then as a continuous positive function, $r$ is uniformly positive and there exists a neighborhood $W_\delta := \{x \in \mathbb{R}^d : \mathrm{dist}(x, \mathcal{M}) < \delta\}$ on which the unique closest point projection is defined. Additionally, we find that

$$\big\|\Psi(\hat{y}, s) - \pi \circ \Psi(\hat{y}, s)\big\| = \left\|\sum_{i=k+1}^{d} s_i A_i(\Psi(\hat{y}, 0))\right\| = \|s\|$$

since the vector fields $A_i$ are orthonormal.

In Lemma 7, we require $\pi(x(t))$ to be $C^2$-smooth if $x(t)$ is $C^2$-smooth due to the technicalities of the proof. For this reason, we make the assumption that $\mathcal{M}$ is a $C^3$-manifold to ensure that $\pi$ is a $C^2$-map. For the required smoothness of solutions to the heavy ball ODE, it is sufficient to require that $f$ is $C^2$-smooth.

**Step 2. The geometry of $f$.** Since $\mathcal{M}$ is the set of minimizers of $f$, the Hessian $D^2 f(z)$ is positive semi-definite for every $z \in \mathcal{M}$. As $f$ is constant on $\mathcal{M}$, we for every curve $\gamma : (-\varepsilon, \varepsilon) \to \mathcal{M}$ we have

$$0 = \frac{d^2}{dt^2} f(\gamma(t)) = \nabla f(\gamma(t)), \gamma''(t)\rangle + \gamma'(t)^T D^2 f(\gamma(t))\, \gamma'(t) = \gamma'(t)^T D^2 f(\gamma(t))\, \gamma'(t)$$

because $\nabla f \equiv 0$ on the set $\mathcal{M}$ of minimizers of $f$, i.e. $v^T D^2 f(z)v = 0$ for all $z \in \mathcal{M}$ and $v \in T_z \mathcal{M}$. If we assume that $D^2 f(z)$ has rank $d - k$ for all $z \in \mathcal{M}$, then necessarily $v^T D^2 f(z)v > 0$ for all $v$ which are orthogonal to $\mathcal{M}$ or equivalently

$$v^T D^2 f(z)v \geq \lambda(z)\|P_z^\perp v\|^2 \qquad \forall\, z \in \mathcal{M},\ v \in \mathbb{R}^d$$

where $P_z^\perp$ denotes the orthogonal projection onto the orthogonal complement of the tangent space of $\mathcal{M}$ at $z$, i.e.

$$P_z^\perp v = \sum_{i=k+1}^{d} \langle v, A_i(z)\rangle A_i(z).$$

If $\mathcal{M}$ is compact, then the function $\lambda$ is bounded from below by some $\lambda_0 > 0$. Let $\varepsilon = \lambda_0/2$. Using the uniform continuity of $\pi, P$ and $D^2 f$ on a compact set $\overline{W_\delta}$ and choosing $\delta > 0$ suitably small, we find that

$$v^T D^2 f(x)v \geq \frac{\lambda_0}{2}\|P_{\pi(x)}^\perp v\|^2 - \varepsilon\|v\|^2 \geq \qquad \forall\, x \in \overline{W_\delta},\ v \in S^{d-1}$$

since the map from matrix to smallest eigenvalue is continuous on the space of symmetric matrices. In particular, for any fixed $(\hat{y}, s) \in \hat{V} \times S^{d-1}$ we see that the function

$$g : (-\delta, \delta) \to \mathbb{R}, \qquad g(t) = f\big(\Psi(\hat{y}, ts)\big)$$

is $\lambda_0 - \varepsilon$ strongly convex. To see this, abbreviate $v := \sum_{i=k+1}^{d} s^i A_i(\hat{y})$ and compute

$$g''(t) = \frac{d^2}{dt^2} f\left(\Psi(\hat{y}, 0) + tv\right) = v^T D^2 f\left(\Psi(\hat{y}, ts)\right)v \geq (\lambda_0 - \varepsilon)\|v\|^2 = \lambda_0 - \varepsilon$$

since $v \in T_{\Psi(\hat{y},0)}\mathcal{M}^\perp$ and $\|v\| = \|s\| = 1$. Hence

$$f\big(\Psi(\hat{y}, 0)\big) = g(0) \geq g(t) - g'(t)t + \frac{\lambda_0 - \varepsilon}{2}t^2$$

$$= f\big(\Psi(\hat{y}, ts)\big) - t\langle \nabla f(\Psi(\hat{y}, ts)), v\rangle + \frac{\lambda_0 - \varepsilon}{2}\|tv\|^2$$

$$= f\big(\Psi(\hat{y}, ts)\big) + \langle \nabla f(\Psi(\hat{y}, ts)), \Psi(\hat{y}, 0) - \Psi(\hat{y}, ts)\rangle + \frac{\lambda_0 - \varepsilon}{2}\big\|\Psi(\hat{y}, 0) - \Psi(\hat{y}, ts)\big\|^2$$

or in the original coordinates of $W_\delta \subseteq \mathbb{R}^d$:

$$f(\pi(x)) \geq f(x) + \langle \nabla f(x), \pi(x) - x\rangle + \frac{\lambda_0 - \varepsilon}{2}\|x - \pi(x)\|^2.$$

By the same argument with reversed roles for $x, \pi(x)$ we have

$$f(x) \geq f(\pi(x)) + \frac{\lambda_0 - \varepsilon}{2}\|x - \pi(x)\|^2 = \inf f + \frac{\lambda_0 - \varepsilon}{2}\operatorname{dist}(x, \mathcal{M})^2.$$

In particular: $f(x) < \inf f + \alpha$ implies that

$$\operatorname{dist}(x, \mathcal{M}) \leq \sqrt{\frac{2}{\lambda_0 - \varepsilon}\big(f(x) - \inf f\big)} < \sqrt{\frac{2\alpha}{\lambda_0 - \varepsilon}}.$$

Choosing $\alpha$ small enough, we see that the open neighborhood

$$\mathcal{U}_\alpha := \{x : f(x) < \inf f + \alpha\}$$

is a subset of $W_\delta$. Within this neighborhood, the unique closest point projection is therefore well-defined. This concludes the proof of the Lemma and shows that the Assumptions of Section 2.1 are satisfied in this setting. □

*Remark* 17. Controlling the largest eigenvalue of the Hessian rather than the smallest, we see that there exist $0 < \mu < L$ such that

$$\frac{\mu}{2}\operatorname{dist}(x, \mathcal{M})^2 \leq f(x) \leq \frac{L}{2}\operatorname{dist}(x, \mathcal{M})^2$$

in a neighborhood of $\mathcal{U}$. For this reason, we presented Example 3 for context and intuition.

# B  PROOFS OF ACCELERATION IN OPTIMIZATION

## B.1  PROOF OF THEOREMS 6 AND 8: OPTIMIZATION IN CONTINUOUS TIME

We first prove the 'local' convergence statement.

**Theorem 6.** *[Continuous time convergence guarantee] Assume that $f$ satisfies the assumptions of Section 2.1, $\gamma = 2\sqrt{\mu}$ and that $x_0$ satisfies $f(x_0) < \alpha$ where $\alpha$ is as in Section 2.1. Then there exists a unique solution $x(t) = x_t$ of (3) and $f(x_t) < \alpha$ for all $t > 0$. Furthermore, $x$ is $C^2$-smooth and*

$$f(x_t) - \inf_{z \in \mathbb{R}^d} f(z) \leq e^{-\sqrt{\mu}t} \left( f(x_0) - \inf f + \frac{\mu}{2} \operatorname{dist}(x_0, \mathcal{M})^2 \right).$$

*Proof.* **Step 0: Existence and Uniqueness.** Note that a solution to the heavy ball ODE can be obtained as a solution to the ODE system

$$\begin{cases} \dot{x} &= v \\ \dot{v} &= -\gamma v - \nabla f(x). \end{cases}$$

If $\nabla f$ is locally Lipschitz-continuous (for instance if $f$ is $C^2$-smooth), a unique $C^1$-solution $(x, v)$ of the ODE system exists by the Picard-Lindelöff Theorem. Since $\dot{x} = v$ is $C^1$-smooth, we see that the solution $x$ of the heavy ball ODE is $C^2$-smooth.

**Step 1: $x_t$ remains in $\mathcal{U}_\alpha$.** Note that

$$\frac{d}{dt} \left( f(x_t) + \frac{1}{2}\|\dot{x}_t\|^2 \right) = \langle \nabla f(x_t), \dot{x}_t \rangle + \langle \dot{x}_t, \ddot{x}_t \rangle = \langle \ddot{x} + \nabla f(x), \dot{x} \rangle = -2\sqrt{\mu}\,\|\dot{x}_t\|^2 \leq 0,$$

so

$$f(x_t) \leq f(x_t) + \frac{1}{2}\|\dot{x}_t\|^2 \leq f(x_0) + \frac{1}{2}\|\dot{x}_0\|^2 = f(x_0) < \alpha$$

for all $t \geq 0$ since $\dot{x}_0 = 0$, which implies $x_t \in \mathcal{U}_\alpha$ for all $t \geq 0$.

**Step 2: Bounding $f(x_t)$.** Let $z_t := \pi(x_t)$ denote the closest point projection of $x_t$ onto $\mathcal{M}$ and by $\dot{z}_t$ its derivative. Consider the Lyapunov function

$$\mathcal{L}(t) = f(x_t) - f(\pi(x_t)) + \frac{1}{2}\|\dot{x}_t + \sqrt{\mu}(x_t - \pi(x_t))\|^2$$

We will show that $\mathcal{L}'(t) \leq -\sqrt{\mu}\mathcal{L}(t)$ under some neighborhood assumption on $x_t$. Using the heavy ball dynamics and properties of the projection, we can bound $\mathcal{L}'(t)$ as:

$$\begin{aligned}
\mathcal{L}'(t) &= \langle \nabla f(x_t), \dot{x}_t \rangle + \langle \dot{x}_t + \sqrt{\mu}(x_t - \pi(x_t)), \ddot{x}_t + \sqrt{\mu}(\dot{x}_t - \dot{z}_t) \rangle \\
&= \langle \dot{x}_t,\, \nabla f(x_t) + \ddot{x}_t + \sqrt{\mu}\,\dot{x}_t - \sqrt{\mu}\,\dot{z}_t \rangle + \sqrt{\mu} \langle x_t - \pi(x_t),\, \ddot{x}_t + \sqrt{\mu}\,\dot{x}_t - \sqrt{\mu}\dot{z}_t \rangle \\
&= \langle \dot{x}_t, -\sqrt{\mu}\,\dot{x}_t - \sqrt{\mu}\dot{z}_t, \rangle + \sqrt{\mu} \langle x_t - z_t,\, -\sqrt{\mu}\,\dot{x}_t - \nabla f(x_t) - \sqrt{\mu}\,\dot{z}_t \rangle \\
&\leq -\sqrt{\mu}\|\dot{x}_t\|^2 - \mu\langle x_t - z_t, \dot{x}_t \rangle - \sqrt{\mu} \langle \nabla f(x_t),\, x_t - z_t \rangle
\end{aligned}$$

where we used the heavy ball dynamics $\ddot{x}_t = -2\sqrt{\mu}\dot{x}_t - \nabla f(x_t)$ and the geometric properties of the closest point projection:

1.  $-\langle \dot{x}_t,\, \dot{z}_t \rangle \leq 0$ by Lemma 7 and

2.  $\langle x_t - z_t,\, \dot{z}_t \rangle = 0$ since $x_t - z_t$ meets $\mathcal{M}$ orthogonally at $z_t$ and $\dot{z}_t$ is tangent to $\mathcal{M}$ at $z_t$.

Next, using the $\mu$-strong aiming condition, we have

$$\langle \nabla f(x_t), x_t - \pi(x_t) \rangle \geq f(x_t) - f(\pi(x_t)) + \frac{\mu}{2}\|x_t - \pi(x_t)\|^2.$$

Substituting this into the bound on $\mathcal{L}'(t)$ and simplifying gives:

$$\begin{aligned}
\mathcal{L}'(t) &\leq -\sqrt{\mu}\,\|\dot{x}_t\|^2 - \mu\, \langle x_t - \pi(x_t), \dot{x}_t \rangle - \sqrt{\mu} \left( f(x_t) - f(\pi(x_t)) + \frac{\mu}{2}\|x_t - \pi(x_t)\|^2 \right) \\
&= -\sqrt{\mu} \left( f(x_t) - f(\pi(x_t)) + \frac{1}{2}\left\|\dot{x}_t + \sqrt{\mu}(x_t - \pi(x_t))\right\|^2 + \frac{1}{2}\,\|\dot{x}_t\|^2 \right)
\end{aligned}$$

$$\leq -\sqrt{\mu}\,\mathcal{L}(t).$$

Now, we can bound the initial value of the Lyapunov function $\mathcal{L}(0)$ as follows:

$$\mathcal{L}(0) = f(x_0) - \inf_{z \in \mathbb{R}^d} f(z) + \frac{1}{2}\|\dot{x}_0 + \sqrt{\mu}(x_0 - \pi(x_0))\|^2$$

$$= f(x_0) - \inf_{z \in \mathbb{R}^d} f(z) + \frac{\mu}{2}\text{dist}(x_0, \mathcal{M})^2$$

since $\dot{x}_0 = 0$ and $\text{dist}(x, \mathcal{M}) = \|x - \pi(x)\|$. We deduce that

$$\frac{d}{dt}e^{\sqrt{\mu}t}\mathcal{L}(t) = \left(\sqrt{\mu}\mathcal{L}(t) + \mathcal{L}'(t)\right)e^{\sqrt{\mu}t} \leq 0$$

so

$$f(x_t) - \inf f \leq \mathcal{L}(t) \leq e^{-\sqrt{\mu}t}\mathcal{L}(0) = e^{-\sqrt{\mu}t}\left(f(x_0) - \inf f + \frac{\mu}{2}\text{dist}(x_0, \mathcal{M})^2\right) \qquad \square$$

Next, we prove the 'global convergence' statement.

**Theorem 8.** *In addition to the assumptions of Section 2.1, assume that*

1. *for every $R > \inf f$, there exists $L_R > 0$ such that $\nabla f$ is Lipschitz-continuous with Lipschitz constant $L_R$ on $\mathcal{U}_R = \{x : f(x) < R\}$ and*

2. *there exists a value $\delta > 0$ such that $\|\nabla f(x)\|^2 < \delta$ implies that $f(x) - \inf f < \frac{\alpha}{2}$.*

*Then, for any $x_0 \in \mathbb{R}^d$, there exists $T \geq 0$ such that $x(t) \in \mathcal{U}_\alpha$ for all $t \geq T$ and such that $f(x(t)) - \inf f \leq \left(3\alpha/2 + \text{dist}(x_T, \mathcal{M})^2\right)e^{\sqrt{\mu}(T-t)}$ for all $t > T$.*

*Proof.* The idea of the proof is to show that at some large time $T$, the trajectory of $x$ enters the set $\mathcal{U}_\alpha$ with sufficiently low velocity that it gets trapped in $\mathcal{U}_\alpha$. From that point onwards, the proof of Theorem 6 applies with minor modifications.

**Step 1.** Denote by $E(t) = f(x_t) + \frac{1}{2}\|\dot{x}_t\|^2$ the 'total energy' of the curve $x$ at time $t$. As in the proof of Theorem 6, we find that $E'(t) = -2\sqrt{\mu}\|\dot{x}_t\|^2$, so in particular $\|\dot{x}\|^2$ is square integrable in time:

$$\int_0^\infty \|\dot{x}\|^2\,\mathrm{d}t = \frac{E(0) - \lim_{t \to \infty} E(t)}{2\sqrt{\mu}} \leq \frac{f(x_0) - \inf f}{2\sqrt{\mu}}.$$

Recall for future use that $f(x_t) \leq E(t) \leq E(0) = f(x_0)$ for $t \geq 0$.

**Step 2.** In this step, we show that also $\nabla f(x_t)$ is square integrable in time. To do this, we first observe that

$$\int_0^T \|\nabla f(x) + 2\sqrt{\mu}\,\dot{x}\|^2\,\mathrm{d}t = \int_0^T \|\nabla f(x)\|^2 + 4\sqrt{\mu}\langle\nabla f(x), \dot{x}\rangle + \|\dot{x}\|^2\,\mathrm{d}t$$

$$= \int_0^T \|\nabla f(x)\|^2\,\mathrm{d}t + 4\sqrt{\mu}\int_0^T \frac{d}{dt}f(x_t)\,\mathrm{d}t + 4\mu\int_0^T \|\dot{x}\|^2\,\mathrm{d}t.$$

On the other hand, we can write

$$\int_0^T \|\nabla f(x) + 2\sqrt{\mu}\,\dot{x}\|^2\,\mathrm{d}t = -\int_0^T \langle\nabla f(x) + 2\sqrt{\mu}\,\dot{x},\,\ddot{x}\rangle\,\mathrm{d}t$$

$$= -\int_0^T \langle\nabla f(x), \ddot{x}\rangle\,\mathrm{d}t - \sqrt{\mu}\int_0^T \frac{d}{dt}\|\dot{x}\|^2\,\mathrm{d}t$$

$$= -\langle\nabla f(x_T), \dot{x}_T\rangle + \int_0^T \langle D^2 f(x)\dot{x}, \dot{x}\rangle\,\mathrm{d}t - \sqrt{\mu}\|\dot{x}_T\|^2$$

since $\dot{x}_0 = 0$. Overall, we find that

$$\int_0^T \|\nabla f(x)\|^2\,\mathrm{d}t = 4\sqrt{\mu}\left(f(x_0) - f(x_T)\right) + \langle\nabla f(x_T), \dot{x}_T\rangle - \sqrt{\mu}\|\dot{x}_T\|^2$$

$$+ \int_0^T \langle D^2 f(x)\dot{x}, \dot{x}\rangle - 4\mu \|\dot{x}\|^2 \, \mathrm{d}t$$

$$\leq 4\sqrt{\mu}\big(f(x_0) - \inf f\big) + L \int_0^T \|\dot{x}\|^2 \, \mathrm{d}t + \langle \nabla f(x), \dot{x}\rangle(T)$$

where $L = L_{f(x_0)}$ is the Lipschitz-constant of $\nabla f$ on the set $\{x : f(x) < f(x_0)\}$. Note that

$$t \mapsto \langle \nabla f(x), \dot{x}\rangle = \frac{d}{dt} f(x_t)$$

is the derivative of the bounded function $f(x_t) \in [\inf f, f(x_0)]$ and continuous, so there exists a sequence of times $T_n \to \infty$ such that $\langle f(x), \dot{x}\rangle(T_n) \to 0$. Since $\|\nabla f\|^2$ is a non-negative integrand, we can bound

$$\int_0^\infty \|\nabla f(x)\|^2 \, \mathrm{d}t \leq \lim_{n \to \infty} \left( 4\sqrt{\mu}\big(f(x_0) - \inf f\big) + L \int_0^{T_n} \|\dot{x}\|^2 \, \mathrm{d}t + \langle \nabla f(x), \dot{x}\rangle(T_n) \right)$$

$$= 4\sqrt{\mu}\big(f(x_0) - \inf f\big) + L \int_0^\infty \|\dot{x}\|^2 \, \mathrm{d}t < +\infty.$$

**Step 3.** Using Steps 1 and 2, we find that

$$\int_0^T \|\nabla f(x_t)\|^2 + \|\dot{x}_t\|^2 \, \mathrm{d}t < +\infty.$$

In particular, there exists a sequence of times $t_n \to \infty$ such that

$$\|\nabla f(x(t_n))\|^2 + \|\dot{x}(t_n)\|^2 \to 0$$

as $n \to \infty$. We can therefore choose $T > 0$ such that

$$\|\nabla f(x_T)\|^2 + \|\dot{x}_T\|^2 < \min\{\delta, \alpha\}.$$

Then we find that

$$\|\nabla f(x_T)\|^2 < \delta \quad \Rightarrow \quad f(x_T) < \alpha, \qquad \|\dot{x}_T\|^2 < \alpha \quad \Rightarrow \quad E(T) = f(x_T) + \frac{1}{2}\|\dot{x}_T\|^2 < \alpha.$$

In particular, we conclude that $f(x_t) \leq E(t) < \alpha$ for all $t > T$, i.e. $x_t \in \mathcal{U}_\alpha$ for all $t > T$. Thus, by the same argument as Theorem 6, we find that

$$\mathcal{L}(t) := f(x_t) - \inf f + \frac{1}{2}\big\|\dot{x} + \sqrt{\mu}\,(x_t - \pi(x_t))\big\|^2$$

satisfies $\mathcal{L}(t) \leq e^{-\sqrt{\mu}(t-T)} L(T)$ for $t > T$, so

$$f(x_t) - \inf f \leq \mathcal{L}(t) \leq e^{\sqrt{\mu}(T-t)} \left( f(x_T) - \inf f + \frac{1}{2}\big\|\dot{x}_T + \sqrt{\mu}(x_T - \pi(x_T))\big\|^2 \right)$$

$$\leq e^{\sqrt{\mu}(T-t)} \left( f(x_t) - \inf f + \frac{2}{2}\big\{\|\dot{x}_T\|^2 + \|x_T - \pi(x_T)\|^2\big\} \right)$$

$$\leq e^{\sqrt{\mu}(T-t)} \left( \frac{3\alpha}{2} + \mathrm{dist}(x_T, \mathcal{M})^2 \right). \qquad \square$$

*Remark* 18. Obviously, the condition that $\nabla f$ is Lipschitz-continuous on all sublevel sets could easily be relaxed to requiring that the initialization $x_0$ is such that $\nabla f$ is merely Lipschitz-continuous on the set $\{x : f(x) < f(x_0)\}$, or even on every connected component of the set.

## B.2 PROOF OF THEOREM 11: ACCELERATION IN DISCRETE TIME (DETERMINISTIC SETTING)

**Lemma 10.** *Assume that $D^2 f(x) \geq -\varepsilon$ in a ball $B_r(x_0)$. Then*

$$\langle \nabla f(x), x - z\rangle \geq f(x) - f(z) - \frac{\varepsilon}{2}\|x - z\|^2 \qquad \forall\, x, z \in B_r(x_0).$$

*Proof.* The function $f(x) + \frac{\varepsilon}{2}\|x\|^2$ is convex since all eigenvalues of its Hessian $D^2 f + \varepsilon I$ are non-negative, so

$$f(z) + \frac{\varepsilon}{2}\|z\|^2 \geq f(x) + \frac{\varepsilon}{2}\|x\|^2 + \langle \nabla f(x) + \varepsilon x, z - x \rangle$$

which is equivalent to

$$\begin{aligned}
\langle \nabla f(x), x - z \rangle &\geq f(x) - f(z) + \frac{\varepsilon}{2}\|x\|^2 + \varepsilon\langle x, z - x \rangle - \frac{\varepsilon}{2}\|z\|^2 \\
&= f(x) - f(z) + \frac{\varepsilon}{2}\|x\|^2 + \varepsilon\langle x, z \rangle - \varepsilon\|x\|^2 - \frac{\varepsilon}{2}\|z\|^2 \\
&= f(x) - f(z) - \frac{\varepsilon}{2}\|z - x\|^2.
\end{aligned}$$
$\square$

Before proving Theorem 11, we recall a well-known auxiliary result.

**Lemma 19.** *(Gupta et al., 2024, Lemma 13) Assume that $\nabla f$ is Lipschitz-continuous with Lipschitz-constant $L$. Then*

$$f(x - \eta g) \leq f(x) - \eta\langle \nabla f(x), g \rangle + \frac{L\eta^2}{2}\|g\|^2.$$

**Theorem 11.** *Assume that $f$ is $L$-smooth and the sequences $x_n, x_n', v_n$ are generated according to the Nesterov scheme (5) with parameters $\eta \leq 1/L$ and $\rho = (1 - \sqrt{\mu\eta})/(1 + \sqrt{\mu\eta})$. Assume further that there exists an affine linear projection map $\pi(x) = Px + x^*$ such that*

$$\langle \nabla f(x), x - \pi(x) \rangle \geq f(x) - f(\pi(x)) + \frac{\mu}{2}\|x - \pi(x)\|^2. \tag{6}$$

*Finally, assume that for arbitrary $x, v \in \mathbb{R}^d$ we have*

$$\langle \nabla f(x + v), v \rangle \geq f(x + v) - f(x) - \frac{\varepsilon}{2}\|v\|^2 \tag{7}$$

*with some $\varepsilon \leq \sqrt{\mu/\eta}$. Then*

$$f(x_n) - \inf f \leq (1 - \sqrt{\mu\eta})^n \left[ f(x_0) - \inf f + \frac{\mu}{2}\|x_0 - \pi(x_0)\|^2 \right].$$

*Proof.* **Setup.** Denote by $P^\perp = I - P$ the orthogonal projection onto the orthogonal complement of the space which $P$ projects onto. Consider the Lyapunov sequence defined by

$$\mathcal{L}_n = f(x_n) - \inf f + \frac{1}{2}\| P^\perp v_n + \sqrt{\mu}(x_n' - \pi(x_n'))\|^2 + \frac{(1 + \sqrt{\mu\eta})^2}{2(1 - \sqrt{\mu\eta})}\|P v_n\|^2.$$

This is a variation of the usual Lyapunov sequence in which we separately analyze the tangential and normal velocities. Note however that

$$\frac{(1 + \sqrt{\mu\eta})^2}{(1 - \sqrt{\mu\eta})} = 1 + O(\sqrt{\eta}),$$

i.e. if $\eta \to 0$, we recover the Lyapunov function in the continuous time setting where tangential and normal velocity are not separated:

$$\| P^\perp v + \sqrt{\mu}(x - \pi(x))\|^2 + \frac{(1 + \sqrt{\mu\eta})^2}{2(1 - \sqrt{\mu\eta})}\|Pv\|^2 \to \| P^\perp v + \sqrt{\mu}(x - \pi(x))\|^2 + \|Pv\|^2 = \|v + \sqrt{\mu}(x - \pi(x))\|^2$$

as $\eta \to 0$ since the vectors $Pv$ and $P^\perp v + x - \pi(x)$ are orthogonal and $v = Pv + P^\perp v$. We want to show that $\mathcal{L}_{n+1} \leq (1 - \sqrt{\mu\eta})\mathcal{L}_n$. Note that $1 - \sqrt{\mu\eta} \geq 1 - \sqrt{\frac{\mu}{L}} \geq 0$ since $\mu \leq L$ by Lemma 24.

For simplicity, we assume that $x^* = 0$, i.e. $x_n' - \pi(x_n') = x_n' - Px_n' = P^\perp x_n'$.

**Step 1.** Since $f$ is $L$-smooth and $\eta \leq 1/L$, we have

$$f(x_{n+1}) \leq f(x_n') - \left(1 - \frac{L\eta}{2}\right)\eta\|\nabla f(x_n')\|^2 = f(x_n') - \frac{\eta}{2}\|\nabla f(x_n')\|^2$$

by Lemma 19 with $x = x_n'$ and $g = \nabla f(x_n')$.

**Step 2.** We compute

$$
\begin{aligned}
P^\perp v_{n+1} + \sqrt{\mu} P^\perp x'_{n+1} &= P^\perp v_{n+1} + \sqrt{\mu} P^\perp (x'_n + \sqrt{\eta} v_{n+1} - \eta \nabla f(x'_n)) \\
&= (P^\perp + \sqrt{\mu\eta} P^\perp) v_{n+1} + \sqrt{\mu} P^\perp x'_n - \eta \sqrt{\mu}\, P^\perp \nabla f(x'_n) \\
&= \rho(1 + \sqrt{\mu\eta}) P^\perp v_n + \sqrt{\mu} P^\perp x'_n - \sqrt{\eta}\left(\sqrt{\mu\eta} + \rho(1 + \sqrt{\mu}\sqrt{\eta})\right) P^\perp \nabla f(x'_n) \\
&= (1 - \sqrt{\mu\eta}) P^\perp v_n + \sqrt{\mu} P^\perp x'_n - \sqrt{\eta}\, P^\perp \nabla f(x'_n),
\end{aligned}
$$

where we simplify the coefficients in the last step by substituting $\rho = \frac{1 - \sqrt{\mu\eta}}{1 + \sqrt{\mu\eta}}$. So,

$$
\begin{aligned}
\frac{1}{2}\left\| P^\perp v_{n+1} + \sqrt{\mu} P^\perp x'_{n+1} \right\|^2 = {}& \frac{(1 - \sqrt{\mu\eta})^2}{2}\| P^\perp v_n \|^2 + \frac{\mu}{2}\| P^\perp x'_n \|^2 + \frac{\eta}{2}\| P^\perp \nabla f(x'_n) \|^2 \\
&+ \sqrt{\mu}(1 - \sqrt{\mu\eta}) \langle P^\perp v_n,\, P^\perp x'_n \rangle - \sqrt{\mu\eta}\, \langle \nabla f(x'_n),\, P^\perp x'_n \rangle \\
&- \sqrt{\eta}(1 - \sqrt{\mu\eta}) \langle \nabla f(x'_n),\, P^\perp v_n \rangle.
\end{aligned}
$$

Recall that since both $P$ and $P^\perp$ are orthogonal projections, for any $x, y \in \mathbb{R}^d$, $\langle Px, Py \rangle = \langle Px, y \rangle = \langle x, Py \rangle$, and the analogous result holds for $P^\perp$ as well.

**Step 3.** Now we expand the last term in the Lyapunov sequence,

$$
\begin{aligned}
\frac{(1 + \sqrt{\mu\eta})^2}{2(1 - \sqrt{\mu\eta})}\| P v_{n+1} \|^2 &= \frac{(1 + \sqrt{\mu\eta})^2}{2(1 - \sqrt{\mu\eta})} \rho^2 \| P(v_n - \sqrt{\eta} \nabla f(x'_n)) \|^2 \\
&= \frac{1 - \sqrt{\mu\eta}}{2}\left( \| P v_n \|^2 + \eta \| P \nabla f(x'_n) \|^2 - 2\sqrt{\eta} \langle \nabla f(x'_n),\, P v_n \rangle \right).
\end{aligned}
$$

**Step 4.** We add the expressions from the previous steps and use the fact that $P v_n + P^\perp v_n = v_n$ to get

$$
\begin{aligned}
\mathcal{L}_{n+1} = {}& \frac{(1 - \sqrt{\mu\eta})^2}{2}\| P^\perp v_n \|^2 + \frac{\mu}{2}\| P^\perp x'_n \|^2 + \frac{\eta}{2}\| P^\perp \nabla f(x'_n) \|^2 \\
&+ \sqrt{\mu}(1 - \sqrt{\mu\eta}) \langle P^\perp v_n,\, P^\perp x'_n \rangle - \sqrt{\mu\eta}\, \langle \nabla f(x'_n),\, P^\perp x'_n \rangle \\
&- \sqrt{\eta}(1 - \sqrt{\mu\eta}) \langle \nabla f(x'_n),\, v_n \rangle + \frac{1 - \sqrt{\mu\eta}}{2}\| P v_n \|^2 \\
&+ \frac{\eta(1 - \sqrt{\mu\eta})}{2}\| P \nabla f(x'_n) \|^2 + f(x_{n+1}) - \inf f.
\end{aligned}
$$

Using (6) and (7) to bound the inner products $\langle \nabla f(x'_n), v_n \rangle$ and $\langle \nabla f(x'_n), P^\perp x'_n \rangle$,

$$
\begin{aligned}
\mathcal{L}_{n+1} \leq {}& \frac{(1 - \sqrt{\mu\eta})^2}{2}\| P^\perp v_n \|^2 + \frac{\mu}{2}\| P^\perp x'_n \|^2 + \frac{\eta}{2}\| P^\perp \nabla f(x'_n) \|^2 \\
&+ \sqrt{\mu}(1 - \sqrt{\mu\eta}) \langle P^\perp v_n,\, P^\perp x'_n \rangle - \sqrt{\mu\eta}\left( f(x'_n) - \inf f + \frac{\mu}{2}\| P^\perp x'_n \|^2 \right) \\
&- (1 - \sqrt{\mu\eta})\left( f(x'_n) - f(x_n) - \frac{\varepsilon}{2}\| \sqrt{\eta}\, v_n \|^2 \right) + \frac{1 - \sqrt{\mu\eta}}{2}\| P v_n \|^2 \\
&+ \frac{\eta(1 - \sqrt{\mu\eta})}{2}\| P \nabla f(x'_n) \|^2 + f(x'_n) - \frac{\eta}{2}\| \nabla f(x'_n) \|^2 - \inf f.
\end{aligned}
$$

Now, we use Pythagoras theorem, i.e. for all $w \in \mathbb{R}^d$, $\| Pw \|^2 + \| P^\perp w \|^2 = \| Pw + P^\perp w \|^2 = \| w \|^2$, and rearrange some of the terms,

$$
\begin{aligned}
\mathcal{L}_{n+1} \leq {}& \frac{(1 - \sqrt{\mu\eta})^2 + (1 - \sqrt{\mu\eta})\eta\varepsilon}{2}\| P^\perp v_n \|^2 + \frac{(1 - \sqrt{\mu\eta})(1 + \eta\varepsilon)}{2}\| P v_n \|^2 \\
&+ \frac{\mu(1 - \sqrt{\mu\eta})}{2}\| P^\perp x'_n \|^2 + \frac{\eta - \eta}{2}\| \nabla f(x'_n) \|^2 - \frac{\sqrt{\eta}^3 \sqrt{\mu}}{2}\| P \nabla f(x'_n) \|^2 \\
&+ \sqrt{\mu}(1 - \sqrt{\mu\eta}) \langle P^\perp v_n,\, P^\perp x'_n \rangle \\
&+ (1 - \sqrt{\mu\eta} - 1 + \sqrt{\mu\eta}) f(x'_n) + (1 - \sqrt{\mu\eta})(f(x_n) - \inf f).
\end{aligned}
$$

The coefficients of $f(x'_n)$ and $\|\nabla f(x'_n)\|^2$ are zero and the coefficient of $\|P\nabla f(x'_n)\|^2$ is negative, so we can disregard those terms. If $\varepsilon \leq \sqrt{\mu/\eta}$, the coefficient $\|P^\perp v_n\|^2$ can be bounded as

$$\frac{(1-\sqrt{\mu\eta})(1-\sqrt{\mu\eta}+\eta\varepsilon)}{2} \leq \frac{(1-\sqrt{\mu\eta})(1-\sqrt{\mu\eta}+\sqrt{\mu\eta})}{2} \leq \frac{1-\sqrt{\mu\eta}}{2},$$

and the coefficient of $\|P^\perp v_n\|^2$ can be bounded as

$$\frac{(1-\sqrt{\mu\eta})(1+\eta\varepsilon)}{2} \leq \frac{(1-\sqrt{\mu\eta})(1+\sqrt{\mu\eta})}{2} \leq \frac{(1+\sqrt{\mu\eta})^2}{2}.$$

Thus, we conclude that

$$\begin{aligned}
\mathcal{L}_{n+1} \leq &(1-\sqrt{\mu\eta})(f(x_n)-\inf f) + \frac{1-\sqrt{\mu\eta}}{2}\|P^\perp v_n\|^2 \\
&+ \frac{\mu(1-\sqrt{\mu\eta})}{2}\|P^\perp x'_n\|^2 + \sqrt{\mu}(1-\sqrt{\mu\eta})\langle P^\perp v_n, P^\perp x'_n\rangle \\
&+ \frac{(1+\sqrt{\mu\eta})^2}{2}\|Pv_n\|^2 \\
= &(1-\sqrt{\mu\eta})\mathcal{L}_n.
\end{aligned}$$

Consequently,

$$f(x_n)-\inf f \leq \mathcal{L}_n \leq (1-\sqrt{\mu\eta})^n L_0 = (1-\sqrt{\mu\eta})^n\left[f(x_0)-\inf f + \frac{\mu}{2}\|P^\perp x_0\|^2\right]. \quad \square$$

### B.3 PROOF OF THEOREMS 13 AND 14: ACCELERATION WITH ADDITIVE NOISE

The proof of Theorem 13 mimics that of Theorem 11 with minor modifications to account for stochastic gradient estimates. Since $\omega_n$ is stochastically independent of anything which has happened in the algorithm before, in particular of $x'_n, v_n$, we have the following.

**Lemma 20.** *For all $n \in \mathbb{N}$, we have*

$$\begin{aligned}
\mathbb{E}\big[\langle g(x'_n,\omega_n), \nabla f(x'_n)\rangle\big] &= \mathbb{E}\big[\|\nabla f(x'_n)\|^2\big] \\
\mathbb{E}\big[\langle g(x'_n,\omega_n), v_n\rangle\big] &= \mathbb{E}\big[\langle \nabla f(x'_n), v_n\rangle\big] \\
\mathbb{E}\big[\langle g(x'_n,\omega_n), P^\perp x'_n\rangle\big] &= \mathbb{E}\big[\langle \nabla f(x'_n), P^\perp x'_n\rangle\big] \\
\mathbb{E}\big[\|g_n\|^2\big] &= \mathbb{E}\big[\|\nabla f(x'_n)\|^2\big] + \mathbb{E}\big[\|g_n - \nabla f(x'_n)\|^2\big]
\end{aligned}$$

*where the expectations are taken over the (potentially random) initial condition $x_0$ as well as the random coefficients $\omega_0, \ldots, \omega_n$ which govern the gradient estimates.*

A proof can be found in (Gupta et al., 2024, Lemma 15). The second identity, which is not included therein, can be proved analogously. As an application, we prove a stochastic analogue of Lemma 19.

**Lemma 21.** *Assume that $f$ is $L$-smooth and $\mathbb{E}\big[\|g(x,\omega)-\nabla f(x)\|^2\big] \leq \sigma_a^2 + \sigma_m^2\|\nabla f(x)\|^2$ for all $x \in \mathbb{R}^d$. Then, if $x, \omega$ are independent random variables, we have*

$$\mathbb{E}_{(x,\omega)}\big[f(x-\eta g(x,\omega))\big] \leq \mathbb{E}\big[f(x)\big] - \left(1 - \frac{L(1+\sigma^2)\eta}{2}\right)\eta\,\mathbb{E}\big[\|\nabla f(x)\|^2\big] + \frac{L\eta^2}{2}\sigma_a^2.$$

*Proof.* Recall that for any $x, \eta, g$, the following holds

$$f(x-\eta g) \leq f(x) - \eta\langle \nabla f(x), g\rangle + \frac{L\eta^2}{2}\|g\|^2$$

by Lemma 19. We now assume that $g = g(x,\omega)$ is a random estimator for $\nabla f(x)$, where $x$ may be random, but $\omega$ is independent of $x$. Then, by Lemma 20, we have

$$\begin{aligned}
\mathbb{E}\big[f(x-\eta g)\big] &\leq \mathbb{E}\big[f(x)\big] - \eta\,\mathbb{E}\big[\|\nabla f(x)\|^2\big] + \frac{L\eta^2}{2}\mathbb{E}\big[\|g\|^2\big] \\
&= \mathbb{E}\big[f(x)\big] - \eta\,\mathbb{E}\big[\|\nabla f(x)\|^2\big] + \frac{L\eta^2}{2}\big\{\mathbb{E}\big[\|\nabla f(x)\|^2\big] + \mathbb{E}\big[\|g - \nabla f(x)\|^2\big]\big\}
\end{aligned}$$

$$\leq \mathbb{E}\big[f(x)\big] - \left(1 - \frac{L\eta}{2}\right)\eta\,\mathbb{E}\big[\|\nabla f(x)\|^2\big] + \frac{L\eta^2}{2}\big\{\sigma_a^2 + \sigma_m^2\,\mathbb{E}\big[\|\nabla f(x)\|^2\big]$$

$$= \mathbb{E}\big[f(x)\big] - \left(1 - \frac{L(1+\sigma_m^2)\eta}{2}\right)\eta\,\mathbb{E}\big[\|\nabla f(x)\|^2\big] + \frac{L\eta^2}{2}\,\sigma_a^2. \qquad \square$$

Finally, we provide an auxiliary result to resolve a recursion.

**Lemma 22.** *Assume a sequence $x_n$ satisfies the recursive estimate $x_{n+1} \leq a x_n + b$ for $a \in (0,1)$ and $b \geq 0$. Then*

$$x_n \leq a^n x_0 + \frac{b}{1-a}.$$

*Proof.* Consider the sequence $y_n := x_n + \frac{b}{a-1}$. Then

$$y_{n+1} = x_{n+1} + \frac{b}{a-1} \leq a x_n + b + \frac{b}{a-1} = a x_n + \frac{a-1}{a-1}b + \frac{b}{a-1} = a x_n + \frac{ab}{a-1}$$

$$= a\left(x_n + \frac{b}{a-1}\right) = a y_n.$$

In particular, $y_n \leq a^n y_0$, so

$$x_n = y_n + \frac{b}{1-a} \leq a^n\left(x_0 + \frac{b}{a-1}\right) + \frac{b}{1-a}.$$

If $b > 0$, the estimate follows since $b/(a-1) < 0$. $\qquad \square$

We now prove the first main result of this section.

**Theorem 13.** *[Acceleration with additive noise] Assume that $f, P$ are as in Theorem 11 and that the $g$ satisfies (8) and (9) with $\sigma_m = 0$. Assume that the sequences $x_n, x_n', v_n$ are generated by the scheme (5) for parameters $\eta \leq 1/L$ and $\rho = \frac{1-\sqrt{\mu\eta}}{1+\sqrt{\mu\eta}}$, but with the stochastic gradient estimates $g(x_n', \omega_n)$ with independently identically distributed $\omega_n$ in place of $\nabla f(x_n')$. Then*

$$\mathbb{E}[f(x_n) - \inf f] \leq (1 - \sqrt{\mu\eta})^n \left[f(x_0) - \inf f + \frac{\mu}{2}\|x_0 - \pi(x_0)\|^2\right] + \frac{\sigma_a^2\sqrt{\eta}}{\sqrt{\mu}}.$$

*Proof.* The proof is mostly identical to that of Theorem 11 with minor modifications: In Step 1, we obtain the estimate

$$\mathbb{E}[f(x_{n+1})] \leq \mathbb{E}[f(x_n')] - \frac{\eta}{2}\,\mathbb{E}\big[\|\nabla f(x_n')\|^2\big] + \frac{L\sigma_a^2\eta^2}{2}$$

by Lemma 21. In the quadratic terms in steps 2 and 3, we need to take the expectation of $g_n$ rather than $\nabla f(x_n')$. By construction, the terms involving $\mathbb{E}[\|\nabla f(x_n')\|^2]$ still balance with the same parameters, leading to an additional contribution of $\eta\,\sigma_a^2$ due to the stochastic gradient estimates. Since $\eta \leq 1/L$, we can bound $L\eta^2 \leq \eta$ so overall we obtain the estimate

$$\mathcal{L}_{n+1} \leq (1 - \sqrt{\mu\eta})\mathcal{L}_n + \sigma_a^2\eta.$$

in place of $\mathcal{L}_{n+1} \leq (1 - \sqrt{\mu\eta})\mathcal{L}_n$ since all other terms are identical under the expectation using Lemma 20. The claim now follows from Lemma 22. $\qquad \square$

For readers who are looking for a more detailed proof, we note that Theorem 13 is a special case of Theorem 15 with $\sigma_m = 0$, where we provide a full proof.

In the same spirit, we sketch the proof of Theorem 14. Let us recall the statement.

**Theorem 14.** *[Additive noise and decreasing step size] Assume that $f, g$ are as in Theorem 13 and that the sequences $x_n, x_n', \rho_n$ are generated by the scheme*

$$x_n' = x_n + \sqrt{\eta_{n-1}}v_n, \qquad x_{n+1} = x_n' - \eta_n g_n, \qquad v_{n+1} = \rho_n(v_n - \sqrt{\eta_n}g_n)$$

*for parameters $\eta_n = \frac{\mu}{(n+\sqrt{L\mu}+1)^2}$, $\rho_n = \frac{1-\sqrt{\mu\eta_n}}{1+\sqrt{\mu\eta_n}}$. If $\varepsilon \leq \sqrt{\mu/\eta_0} = \mu + \sqrt{L\mu}$, then*

$$\mathbb{E}[f(x_n) - \inf f] \leq \frac{\sqrt{\frac{L}{\mu}}\,\mathbb{E}\left[f(x_0) - \inf f + \frac{1}{2}\|x_0 - \pi(x_0)\|^2\right] + \frac{\sigma_a^2}{\mu}\log\left(1 + n\sqrt{\mu/L}\right)}{n + \sqrt{L/\mu}}.$$

*Proof.* Note that in the proof of Theorem 15, we build on the relationships

$$f(x_{n+1}) \le f(x'_n) - \frac{\eta_n}{2} \|\nabla f(x'_n)\|^2$$

$$x'_{n+1} = x'_n - \eta_n g_n + \sqrt{\alpha_n} v_n$$

$$v_{n+1} = \rho_n(v_n - \sqrt{\alpha_n} g_n)$$

and do not enter further into the recursion. We additionally note that if $\eta_n$ is a *monotone decreasing* sequence, then the sequence $\lambda_{n+1} := \frac{(1+\sqrt{\mu\eta_n})^2}{1-\sqrt{\mu\eta_n}}$ is also monotone decreasing. Hence, if $\eta_n \le \frac{1}{L(1+\sigma_m^2)}$ for all $n \in \mathbb{N}$, then by the same proof as Theorem 13, the sequence

$$\mathcal{L}_{n+1} := f(x_n) - \inf f + \frac{1}{2}\|P^\perp v_n + \sqrt{\mu} P^\perp x'_n\|^2 + \frac{\lambda_n}{2}\|P v_n\|^2$$

satisfies

$$\mathcal{L}_n \le \left(1 - \sqrt{\mu\eta_n}\right)\left\{ f(x_n) - \inf f + \frac{1}{2}\|P^\perp v_n + \sqrt{\mu} P^\perp x'_n\|^2 + \frac{\lambda_{n+1}}{2}\|P v_n\|^2 \right\} + \sigma_a^2 \eta_n$$

$$\le \left(1 - \sqrt{\mu\eta_n}\right)\mathcal{L}_n + \sigma_a^2 \eta_n$$

if the parameters are chosen as in the theorem statement. If specifically $\eta_n = \frac{1}{\mu(n+n_0+1)^2}$ for $n_0 = \sqrt{L/\mu}$, then

$$\mathcal{L}_{n+1} \le \left(1 - \frac{1}{n+n_0+1}\right)\mathcal{L}_n + \frac{\sigma_a^2}{\mu(n+n_0+1)^2} = \frac{n+n_0}{n+n_0+1}\mathcal{L}_n + \frac{\sigma_a^2}{\mu(n+n_0+1)^2}.$$

Thus the sequence $z_n := (n+n_0)\mathcal{L}_n$ satisfies the relation

$$z_{n+1} = (n+n_0+1)\mathcal{L}_{n+1} \le (n+n_0)\mathcal{L}_n + \frac{\sigma_a^2}{\mu(n+n_0+1)} = z_n + \frac{\sigma_a^2}{\mu(n+n_0+1)}$$

so

$$z_n = z_0 + \sum_{k=1}^{n}(z_k - z_{k-1}) \le z_0 + \sum_{k=0}^{n-1} \frac{\sigma_a^2}{\mu(k+n_0+1)} \le n_0 \mathcal{L}_0 + \frac{\sigma_a^2}{\mu} \int_0^{n-1} \frac{1}{n_0+t} \, dt$$

$$\le n_0 \mathcal{L}_0 + \frac{\sigma_a^2}{\mu} \log\left(1 + \frac{n-1}{n_0}\right).$$

Overall, we find that

$$\mathbb{E}\big[f(x_n) - \inf f\big] \le \mathcal{L}_n = \frac{z_n}{n+n_0}$$

$$\le \frac{\sqrt{\frac{L}{\mu}}\,\mathbb{E}\big[f(x_0) - \inf f + \frac{1}{2}\|x_0 - \pi(x_0)\|^2\big] + \frac{\sigma_a^2}{\mu}\log\left(1 + \frac{n-1}{n_0}\right)}{n+n_0}.$$

$\square$

### B.4 PROOF OF THEOREM 15: ACCELERATION WITH BOTH ADDITIVE AND MULTIPLICATIVE NOISE

In the strongly convex setting, Gupta et al. (2024) state a version of Theorem 15 in slightly greater generality in terms of choosing variables. The same more general proof goes through also here after we account for the tangential and normal components of the velocity as in the proof of Theorem 11.

**Theorem 15.** *[Additive and multiplicative noise] Assume that $f, P, x^*$ are as in Theorem 11 and that $g$ is a family of gradient estimators such that (8) and (9) hold for some $\sigma_a, \sigma_m \ge 0$. Assume that the sequences $x_n, x'_n, v_n$ are generated by the AGNES scheme (10) with parameters*

$$0 < \eta \le \frac{1}{L(1+\sigma_m^2)}, \qquad \rho = \frac{1 - \sqrt{\frac{\mu\eta}{1+\sigma_m^2}}}{1 + \sqrt{\frac{\mu\eta}{1+\sigma_m^2}}}, \qquad \alpha = \frac{1 - \sqrt{\mu(1+\sigma_m^2)\eta}}{1 - \sqrt{\mu(1+\sigma_m^2)\eta} + \sigma_m^2}\,\eta.$$

*Then, if $\varepsilon < \sqrt{\mu(1+\sigma_m^2)/\eta}$, we have*

$$\mathbb{E}\big[f(x_n) - \inf f\big] \le \left(1 - \sqrt{\frac{\mu\eta}{1+\sigma_m^2}}\right)^n \mathbb{E}\left[f(x_0) - \inf f + \frac{\mu}{2}\|x_0 - \pi(x_0)\|^2\right] + \frac{\sigma_a^2\sqrt{\eta}}{\sqrt{\mu(1+\sigma_m^2)}}.$$

*Proof.* **Setup.** Mimicking the proof of Theorem 11, consider the sequence

$$\mathcal{L}_n = \mathbb{E}\big[f(x_n) - f(x^*)\big] + \frac{1}{2}\,\mathbb{E}\big[\|b\,P^\perp v_n + a(x_n' - \pi(x_n'))\|^2\big] + \frac{\lambda}{2}\,\mathbb{E}\big[\|Pv_n\|^2\big]$$

for constants

$$b = \sqrt{\frac{(1 + \sigma_m^2)\alpha}{\eta}}, \qquad \lambda = \frac{(b + \sqrt{\mu}\gamma)^2}{b - \sqrt{\mu}\,\gamma}\,\frac{\gamma}{\sqrt{\alpha}} \qquad \text{where} \quad \gamma = \sqrt{\mu}(\eta - \alpha) + b\sqrt{\alpha}.$$

The constants will be motivated below where they are introduced. Note that we have $\eta = \alpha$ if $\sigma_m = 0$ and thus $b = 1$ and $\gamma = \sqrt{\alpha}$, recovering the situation of Theorem 13. We want to show that $\mathcal{L}_{n+1} \leq (1 - \sqrt{\mu}\,\sqrt{\alpha}/b)\mathcal{L}_n$. For simplicity, we again assume without loss of generality that $\pi(x) = Px$, i.e. $x^* = 0$ throughout the proof.

**Step 1.** Consider the first term first. Note that

$$\mathbb{E}\big[f(x_{n+1})\big] = \mathbb{E}\big[f(x_n' - \eta g_n)\big] \leq \mathbb{E}\big[f(x_n')\big] - \left(1 - \frac{L(1 + \sigma^2)}{2}\eta\right)\eta\,\mathbb{E}\big[\|\nabla f(x_n')\|^2\big] + \frac{L\eta^2}{2}\sigma_a^2$$

$$\leq \mathbb{E}\big[f(x_n')\big] - \frac{\eta}{2}\,\mathbb{E}\big[\|\nabla f(x_n')\|^2\big] + \frac{L\eta^2}{2}\sigma_a^2$$

if $\eta \leq \frac{1}{L(1+\sigma^2)}$ by Lemma 21.

**Step 2.** We now turn to the second term and use the definition of $x_{n+1}'$ from (10),

$$bP^\perp v_{n+1} + \sqrt{\mu}P^\perp x_{n+1}' = bP^\perp v_{n+1} + \sqrt{\mu}P^\perp(x_n' + \sqrt{\alpha}\,v_n - \eta g_n)$$

$$= (b + \sqrt{\mu\alpha})\rho P^\perp\big(v_n - \sqrt{\alpha}g_n\big) + \sqrt{\mu}P^\perp x_n' - \sqrt{\mu}\eta P^\perp g_n$$

$$= (b + \sqrt{\mu\alpha})\rho P^\perp v_n + \sqrt{\mu}P^\perp x_n' - \big(\sqrt{\mu}\eta + \rho(b + \sqrt{\mu\alpha})\sqrt{\alpha}\big)P^\perp g_n.$$

In analogy to the proof of Theorem 11, we have $\rho = \frac{b - \sqrt{\mu\alpha}}{b + \sqrt{\mu\alpha}}$, so

$$bP^\perp v_{n+1} + \sqrt{\mu}P^\perp x_{n+1}' = (b - \sqrt{\mu\alpha})P^\perp v_n + \sqrt{\mu}P^\perp x_n' - \big(\sqrt{\mu}\eta + (b - \sqrt{\mu\alpha})\sqrt{\alpha}\big)g_n.$$

In the deterministic case where $\eta = \alpha$, the coefficient $\sqrt{\mu}(\eta - \alpha) + b\sqrt{\alpha}$ of $g_n$ simplified to $b\sqrt{\alpha}$. It does not in this more general setting anymore, so we introduce a new notation: $\gamma = \sqrt{\mu}(\eta - \alpha) + b\sqrt{\alpha}$.

Taking expectation of the square, we find that

$$\mathbb{E}\left[\big\|bP^\perp v_{n+1} + \sqrt{\mu}\,P^\perp x_{n+1}'\big\|^2\right]$$

$$= (b - \sqrt{\mu\alpha})^2\,\mathbb{E}\big[\|P^\perp v_n\|^2\big] + 2\sqrt{\mu}(b - \sqrt{\mu\alpha})\,\mathbb{E}\big[\langle P^\perp v_n, P^\perp x_n'\rangle\big] + a^2\mathbb{E}\big[\|P^\perp x_n'\|^2\big]$$

$$- 2(b - \sqrt{\mu\alpha})\gamma\,\mathbb{E}\big[\langle g_n, P^\perp v_n\rangle\big] - 2\sqrt{\mu}\gamma\,\mathbb{E}\big[\langle g_n, P^\perp x_n'\rangle\big] + \gamma^2\,\mathbb{E}\big[\|P^\perp g_n\|^2\big]$$

$$= (b - \sqrt{\mu\alpha})^2\,\mathbb{E}\big[\|P^\perp v_n\|^2\big] + 2\sqrt{\mu}(b - \sqrt{\mu\alpha})\,\mathbb{E}\big[\langle P^\perp v_n, P^\perp x_n'\rangle\big] + \mu\mathbb{E}\big[\|P^\perp x_n'\|^2\big]$$

$$- 2(b - \sqrt{\mu\alpha})\gamma\,\mathbb{E}\big[\langle \nabla f(x_n'), P^\perp v_n\rangle\big] - 2\sqrt{\mu}\gamma\,\mathbb{E}\big[\langle \nabla f(x_n'), P^\perp x_n'\rangle\big] + \gamma^2\,\mathbb{E}\big[\|P^\perp g_n\|^2\big].$$

**Step 3.** We now consider the third term

$$\lambda\,\mathbb{E}\big[\|Pv_{n+1}\|^2\big] = \lambda\rho^2\,\mathbb{E}\big[\|P(v_n - \sqrt{\alpha}g_n)\|^2\big]$$

$$= \lambda\rho^2\,\mathbb{E}\big[\|Pv_n\|^2 + \alpha\|Pg_n\|^2 - 2\sqrt{\alpha}\langle g_n, Pv_n\rangle\big]$$

$$= \lambda\rho^2\,\mathbb{E}\big[\|Pv_n\|^2 + \alpha\|Pg_n\|^2 - 2\sqrt{\alpha}\langle \nabla f(x_n'), Pv_n\rangle\big].$$

**Step 4.** We now add the estimates of steps 2 and 3, with

$$\lambda = \frac{(b - \sqrt{\mu\alpha})\gamma}{\rho^2\sqrt{\alpha}} = \frac{(b + \sqrt{\mu\alpha})^2}{b - \sqrt{\mu\alpha}}\,\frac{\gamma}{\sqrt{\alpha}}$$

such that the coefficients $-2(b - \sqrt{\mu\alpha})\gamma$ of $\mathbb{E}[\langle \nabla f(x_n'), P^\perp v_n\rangle]$ and $-2\lambda\rho^2\sqrt{\alpha}$ of $\mathbb{E}[\langle \nabla f(x_n'), Pv_n\rangle]$ coincide. Note that in the deterministic case $\gamma = b\sqrt{\alpha} = \sqrt{\alpha}$ and we recover the coefficient chosen in the proof of Theorem 11.

$$\mathbb{E}\big[\|bP^\perp v_{n+1} + \sqrt{\mu}P^\perp x_{n+1}'\|^2 + \lambda\|Pv_{n+1}\|^2\big]$$

$$= (b - \sqrt{\mu\alpha})^2 \, \mathbb{E}\big[\|P^\perp v_n\|^2\big] + 2\sqrt{\mu}(b - \sqrt{\mu\alpha}) \, \mathbb{E}\big[\langle P^\perp v_n, P^\perp x_n'\rangle\big] + \mu\mathbb{E}\big[\|P^\perp x_n'\|^2\big]$$

$$- 2(b - \sqrt{\mu\alpha})\gamma \, \mathbb{E}\big[\langle \nabla f(x_n'), v_n\rangle\big] - 2\sqrt{\mu}\gamma \, \mathbb{E}\big[\langle \nabla f(x_n'), P^\perp x_n'\rangle\big] + \frac{(b - \sqrt{\mu\alpha})\gamma}{\sqrt{\alpha}} \, \mathbb{E}\big[\|P v_n\|^2\big]$$

$$+ \gamma^2 \, \mathbb{E}\big[\|P^\perp g_n\|^2\big] + (b - \sqrt{\mu\alpha})\gamma\sqrt{\alpha} \, \mathbb{E}\big[\|P g_n\|^2\big].$$

We note that the coefficient of the norm of the tangential gradient is $(b - \sqrt{\mu\alpha})\sqrt{\alpha}\,\gamma \le \gamma^2$ by the definition of $\gamma = (b - \sqrt{\mu\alpha})\sqrt{\alpha} + \sqrt{\mu}\eta$. Next, we combine this estimate with the bound on $f(x_{n+1})$ from Step 1 and use the geometric conditions (6) and (7) on $f$ to control the inner products of $\nabla f(x_n')$ with $v_n$ and $P^\perp x_n'$ in the previous expression as well as the variance bound (9) for the gradient estimates:

$$\mathcal{L}_{n+1} \le \mathbb{E}\big[f(x_n') - \inf f\big] - \frac{\eta}{2} \, \mathbb{E}\big[\|\nabla f(x_n')\|^2\big] + \frac{\gamma^2}{2}\mathbb{E}\big[\|g_n\|^2\big] + \frac{L\eta^2}{2}\,\sigma_a^2$$

$$+ \frac{(b - \sqrt{\mu\alpha})^2}{2} \, \mathbb{E}\big[\|P^\perp v_n\|^2\big] + \sqrt{\mu}(b - \sqrt{\mu\alpha}) \, \mathbb{E}\big[\langle P^\perp v_n, P^\perp x_n'\rangle\big] + \frac{\mu}{2}\mathbb{E}\big[\|P^\perp x_n'\|^2\big]$$

$$- (b - \sqrt{\mu\alpha})\frac{\gamma}{\sqrt{\alpha}} \, \mathbb{E}\left[f(x_n') - f(x_n) - \frac{\varepsilon\alpha}{2}\|v_n\|^2\right]$$

$$- \sqrt{\mu}\gamma \, \mathbb{E}\left[f(x_n') - \inf f + \frac{\mu}{2}\|P^\perp x_n'\|^2\right] + \frac{(b - \sqrt{\mu\alpha})\gamma}{2\sqrt{\alpha}} \, \mathbb{E}\big[\|P v_n\|^2\big]$$

$$= \left(1 - (b - \sqrt{\mu\alpha})\frac{\gamma}{\sqrt{\alpha}} - \sqrt{\mu}\gamma\right) \mathbb{E}\big[f(x_n')\big] - (1 - \sqrt{\mu}\gamma)\inf f + (b - \sqrt{\mu\alpha})\frac{\gamma}{\sqrt{\alpha}} \, \mathbb{E}[f(x_n)]$$

$$+ \frac{\gamma^2(1 + \sigma_m^2) - \eta}{2} \, \mathbb{E}\big[\|\nabla f(x_n')\|^2\big] + \frac{L\eta^2 + \gamma^2}{2}\sigma_a^2$$

$$+ \frac{(b - \sqrt{\mu\alpha})^2 + (b - \sqrt{\mu\alpha})\varepsilon\gamma\sqrt{\alpha}}{2} \, \mathbb{E}\big[\|P v_n\|^2\big] + \sqrt{\mu}(b - \sqrt{\mu\alpha})\mathbb{E}\big[\langle P^\perp x_n', P^\perp v_n\rangle\big]$$

$$+ \frac{\mu - \sqrt{\mu}\gamma\mu}{2} \, \mathbb{E}\big[\|P^\perp x_n'\|^2\big] + \frac{(b - \sqrt{\mu\alpha})\gamma(1 + \varepsilon\alpha)}{2\sqrt{\alpha}} \, \mathbb{E}\big[\|P v_n\|^2\big].$$

**Step 5.** Recall that

$$\alpha = \frac{1 - \sqrt{\mu\eta(1 + \sigma_m^2)}}{1 - \sqrt{\mu(1 + \sigma_m^2)\eta} + \sigma_m^2}\,\eta, \qquad b = \sqrt{\frac{(1 + \sigma_m^2)\alpha}{\eta}}$$

and thus

$$\rho = \frac{b - \sqrt{\mu\alpha}}{b + \sqrt{\mu\alpha}} = \frac{\sqrt{(1 + \sigma_m^2)\alpha/\eta} - \sqrt{\mu\alpha}}{\sqrt{(1 + \sigma_m^2)\alpha/\eta} + \sqrt{\mu\alpha}} = \frac{1 - \sqrt{\frac{\mu\eta}{1 + \sigma_m^2}}}{1 + \sqrt{\frac{\mu\eta}{1 + \sigma_m^2}}}$$

as desired.

Let us verify that $\gamma = \sqrt{\alpha}/b$. This is equivalent to

$$\sqrt{\mu}(\eta - \alpha) + \sqrt{\frac{1 + \sigma_m^2}{\eta}}\alpha - \sqrt{\frac{\eta}{1 + \sigma_m^2}} = \left(\sqrt{\frac{1 + \sigma_m^2}{\eta}} - \sqrt{\mu}\right)\alpha + \left(\sqrt{\mu}\eta - \sqrt{\frac{\eta}{1 + \sigma_m^2}}\right) = 0$$

i.e. to the choice

$$\alpha = \frac{\sqrt{\mu}\eta - \sqrt{\frac{\eta}{1 + \sigma_m^2}}}{\sqrt{\mu} - \sqrt{\frac{1 + \sigma_m^2}{\eta}}} = \frac{\sqrt{\mu}\eta - \frac{1}{\sqrt{1 + \sigma_m^2}}}{\sqrt{\mu}\eta - \sqrt{1 + \sigma_m^2}}\,\eta = \frac{1 - \sqrt{\mu\eta(1 + \sigma_m^2)}}{1 + \sigma_m^2 - \sqrt{\mu\eta(1 + \sigma_m^2)}}\,\eta$$

which we made above. In particular, we find that

$$(b - \sqrt{\mu\alpha})\frac{\gamma}{\sqrt{\alpha}} + \sqrt{\mu}\gamma = \left(\frac{b}{\sqrt{\alpha}} - \sqrt{\mu} + \sqrt{\mu}\right)\gamma = \frac{b\gamma}{\sqrt{\alpha}} = 1$$

and therefore

$$\left(1 - (b - \sqrt{\mu\alpha})\frac{\gamma}{\sqrt{\alpha}} - \sqrt{\mu}\gamma\right) \mathbb{E}\big[f(x_n')\big] + (1 - \sqrt{\mu}\gamma)\inf f + (b - \sqrt{\mu\alpha})\frac{\gamma}{\sqrt{\alpha}} \, \mathbb{E}[f(x_n)]$$

$$= (1 - \sqrt{\mu}\gamma) \, \mathbb{E}\big[f(x_n) - \inf f\big].$$

In the coefficient of $\mathbb{E}\big[\|\nabla f(x'_n)\|^2\big]$, we have the cancellations

$$(1 + \sigma_m^2)\gamma^2 - \eta = (1 + \sigma_m^2)\frac{\alpha}{b^2} - \eta = \frac{\eta}{(1 + \sigma_m^2)\alpha}\alpha - \eta = 0.$$

By the same analysis, the coefficient of additive noise is

$$L\eta^2 + \gamma^2 = L\eta^2 + \frac{\eta}{1 + \sigma_m^2},$$

so overall

$$\mathcal{L}_{n+1} \le (1 - \sqrt{\mu}\,\gamma) \, \mathbb{E}\big[f(x_n) - \inf f\big] + \frac{L\eta^2 + \frac{\eta}{1+\sigma_m^2}}{2}\sigma_a^2$$
$$+ \frac{(b - \sqrt{\mu\alpha})^2 + (b - \sqrt{\mu\alpha})\varepsilon\gamma\sqrt{\alpha}}{2}\mathbb{E}\big[\|Pv_n\|^2\big] + \sqrt{\mu}(b - \sqrt{\mu\alpha})\mathbb{E}\big[\langle P^{\perp}x'_n, P^{\perp}v_n\rangle\big]$$
$$+ \frac{a^2 - \sqrt{\mu}\gamma\mu}{2}\mathbb{E}\big[\|P^{\perp}x'_n\|^2\big] + \frac{(b - \sqrt{\mu\alpha})\gamma(1 + \varepsilon\sqrt{\alpha})}{2\sqrt{\alpha}}\mathbb{E}\big[\|Pv_n\|^2\big].$$

We now analyze the terms corresponding to the quadratic terms. Using $\gamma = \sqrt{\alpha}/b$, we obtain

$$\frac{(b - \sqrt{\mu\alpha})^2 + (b - \sqrt{\mu\alpha})\varepsilon\gamma\sqrt{\alpha}}{b^2} \le \left(1 - \frac{\sqrt{\mu\alpha}}{b}\right)\left(1 + \frac{(\varepsilon\gamma - \sqrt{\mu})\sqrt{\alpha}}{b}\right)$$
$$= (1 - \sqrt{\mu}\,\gamma)\left(1 + \frac{(\varepsilon\gamma - \sqrt{\mu})\sqrt{\alpha}}{b}\right)$$
$$\le 1 - \sqrt{\mu}\,\gamma$$

for the coefficient of $\mathbb{E}[\|v_n\|^2]$ if $\varepsilon\gamma \le a$, i.e. if $\varepsilon \le \frac{\sqrt{\mu}}{\gamma} = \sqrt{\frac{\mu(1+\sigma_m^2)}{\eta}}$ Analogously, we see that the coefficient of $\mathbb{E}\big[\langle P^{\perp}v_n, P^{\perp}x'_n\rangle\big]$ that

$$\sqrt{\mu}(b - \sqrt{\mu\alpha}) = \sqrt{\mu}b\frac{b - \sqrt{\mu\alpha}}{b} = \sqrt{\mu}b\left(1 - \sqrt{\mu}\frac{\sqrt{\alpha}}{b}\right) = \sqrt{\mu}b(1 - \sqrt{\mu}\,\gamma)$$

and for the coefficient of $\mathbb{E}\big[\|P^{\perp}x'_n\|^2\big]$ that

$$\frac{\mu - \sqrt{\mu}\gamma\mu}{\mu} = 1 - \frac{\gamma\mu}{\sqrt{\mu}} = 1 - \sqrt{\mu}\gamma.$$

Before proceeding to the next term, we observe that

$$\gamma^2 = \frac{\eta}{1 + \sigma^2} \ge \eta\frac{1 - \sqrt{\mu\eta(1 + \sigma^2)}}{1 + \sigma^2 - \sqrt{\mu\eta(1 + \sigma^2)}} = \alpha,$$

and hence $\varepsilon \le \sqrt{\mu}/\gamma \le \sqrt{\mu}\gamma/\alpha$. Finally, we note for the coefficient of $\mathbb{E}[\|Pv_n\|^2]$ that

$$\frac{(b - \sqrt{\mu\alpha})\gamma(1 + \varepsilon\alpha)}{\sqrt{\alpha}\,\lambda} = \frac{(b - \sqrt{\mu\alpha})\gamma(1 + \varepsilon\alpha)(b - \sqrt{\mu\alpha})\sqrt{\alpha}}{\sqrt{\alpha}\,\gamma(b + \sqrt{\mu\alpha})^2}$$
$$= \left(\frac{b - \sqrt{\mu\alpha}}{b + \sqrt{\mu\alpha}}\right)^2(1 + \varepsilon\alpha)$$
$$\le \left(\frac{b - \sqrt{\mu\alpha}}{b}\right)^2(1 + \varepsilon\alpha)$$
$$= (1 - \sqrt{\mu}\gamma)^2(1 + \varepsilon\alpha)$$
$$\le (1 - \sqrt{\mu}\gamma)^2(1 + \sqrt{\mu}\gamma)$$
$$= (1 - \sqrt{\mu}\gamma)(1 - \mu\gamma^2)$$
$$\le 1 - \sqrt{\mu}\gamma$$

as desired. Overall, we find that

$$
\begin{aligned}
\mathcal{L}_{n+1} &\leq (1 - \sqrt{\mu}\gamma)\Bigg\{ \mathbb{E}[f(x_n) - \inf f] + \frac{b^2}{2}\,\mathbb{E}\big[\|P^\perp v_n\|^2\big] + ab\,\mathbb{E}\big[\langle P^\perp v_n, P^\perp x_n'\rangle\big] \\
&\quad + \frac{\mu}{2}\,\mathbb{E}\big[\|P^\perp x_n'\|^2\big] + \frac{\lambda}{2}\,\mathbb{E}\big[\|Pv_n\|^2\big]\Bigg\} + \frac{L\eta^2 + \frac{\eta}{1+\sigma_m^2}}{2}\,\sigma_a^2 \\
&= (1 - \sqrt{\mu}\gamma)\Bigg\{ \mathbb{E}[f(x_n) - \inf f] + \frac{1}{2}\,\mathbb{E}\big[\|P^\perp v_n + P^\perp x_n'\|^2\big] + \frac{\lambda}{2}\,\mathbb{E}\big[\|Pv_n\|^2\big]\Bigg\} \\
&\quad + \frac{L\eta^2 + \frac{\eta}{1+\sigma_m^2}}{2}\,\sigma_a^2 \\
&= (1 - \sqrt{\mu}\gamma)\,\mathcal{L}_n + \frac{L\eta^2 + \frac{\eta}{1+\sigma_m^2}}{2}\,\sigma_a^2.
\end{aligned}
$$

By Lemma 22, we deduce

$$
\mathcal{L}_n \leq (1 - \sqrt{\mu}\gamma)^n \mathcal{L}_0 + \frac{L\eta^2 + \frac{\eta}{1+\sigma_m^2}}{2\sqrt{\mu}\gamma}\,\sigma_a^2 = \frac{L(1+\sigma_m^2)\eta^2 + \eta}{2(1+\sigma_m^2)\sqrt{\mu}\,\sqrt{\eta/(1+\sigma_m^2)}}.
$$

Since $L(1+\sigma_m^2)\eta \leq 1$, we can simplify the noise term to

$$
\frac{L(1+\sigma_m^2)\eta^2 + \eta}{2(1+\sigma_m^2)\sqrt{\mu}\,\sqrt{\eta/(1+\sigma_m^2)}} \leq \frac{\sigma_a^2\sqrt{\eta}}{\sqrt{\mu(1+\sigma_m^2)}}. \qquad \square
$$

## C  A BRIEF COMPARISON OF GEOMETRIC CONDITIONS FOR OPTIMIZATION

### C.1  DEFINITIONS AND ELEMENTARY PROPERTIES

In this section, we compare some common geometric assumptions in optimization theory. Recall the following notions.

**Definition 23.** Let $U \subseteq \mathbb{R}^d$ be an open set. We say that a $C^1$-function $f : U \to \mathbb{R}$

1. is $\gamma$-*quasar convex* if $\operatorname{argmin} f \neq \emptyset$ and if the inequality

$$
\langle \nabla f(x), x - x^* \rangle \geq \gamma\big(f(x) - f(x^*)\big)
$$

   holds for any $x \in U$ and any $x^* \in \operatorname{argmin} f$.

2. star-convex if it is 1-quasar convex.

3. $(\gamma, \mu)$-strongly quasar convex if $\operatorname{argmin} f \neq \emptyset$ and

$$
\langle \nabla f(x), x - x^* \rangle \geq \gamma\left(f(x) - f(x^*) + \frac{\mu}{2}\|x - x^*\|^2\right)
$$

4. *is first order $\mu$-strongly convex* if

$$
f(x) \geq f(z) + \langle \nabla f(z), x - z \rangle + \frac{\mu}{2}\|x - z\|^2 \qquad \forall\, x, z \in U.
$$

5. *satisfies the first order $\mu$-strong aiming condition* if for all $x$ we have

$$
f(\pi(x)) \geq f(x) + \langle \nabla f(x), \pi(x) - x \rangle + \frac{\mu}{2}\|x - \pi(x)\|^2 \qquad \forall\, x \in U
$$

   where

$$
\pi(x) = \operatorname{argmin}\left\{ \|x - z\|^2 : f(z) = \inf_{x' \in U} f(x') \right\}.
$$

   In particular, we assume that the set of minimizers of $f$ is non-empty and that there exists a unique closest point $\pi(x)$ for all $x \in U$.

6. *satisfies a PL condition with PL constant $\mu$ if*

$$\|\nabla f(x)\|^2 \geq 2\mu\big(f(x) - \inf f\big).$$

If $U$ is a convex set, the fourth condition is of course equivalent to regular $\mu$-strong convexity.

**Lemma 24.** 1. *If $f$ is first order $\mu$-strongly convex and has a minimizer in $U$, then $f$ satisfies the $\mu$-strong aiming condition.*

2. *If $f$ satisfies the $\mu$-strong aiming condition, then $f$ satisfies the PL condition with the same constant $\mu$.*

3. *If $f$ is $\mu$-strongly aiming on $\mathcal{U}_\alpha = \{x \in \mathbb{R}^d : f(x) < \alpha\}$, then the line segment connecting $x$ and $\pi(x)$ is contained in $\mathcal{U}_\alpha$ for all $x \in \mathcal{U}_\alpha$.*

4. *If $f$ is $L$-Lipschitz continuous on $\mathcal{U}_\alpha$ and satisfies the PL-inequality with constant $\mu$ on $\mathcal{U}_\alpha$, then $\mu \leq L$.*

5. *If $f$ is $(\gamma, \mu)$-strongly quasar convex, then $\arg\min f$ consists of a single point.*

6. *If $f$ is $\gamma$-quasar convex and $U$ is star-shaped with respect to the minimizer $x^* \in \arg\min f$, then all sub-level sets of $f$ are star-shaped with respect $x^*$. In particular, the set of minimizers is convex.*

If $U = \mathbb{R}^d$, then any strongly convex function $f : U \to \mathbb{R}$ has a minimizer in $U$. On general open sets, this is not guaranteed.

*Proof.* **First claim.** If $f$ is first order $\mu$-strongly convex and has a minimizer in $U$, then the minimizer $x^*$ is unique since for $x \neq x^*$ we have

$$f(x) \geq f(x^*) + \langle \nabla f(x^*), x - x^* \rangle + \frac{\mu}{2} \|x - x^*\|^2 = f(x^*) + \frac{\mu}{2} \|x - x^*\|^2 > f(x^*).$$

In particular, for every $x$ there exists a unique closest minimizer $\pi(x) = x^*$. The strong aiming condition therefore requires the first order convexity condition only for pairs of points $x, x^*$ rather than all points $x, z$.

**Second claim.** This result follows by the same proof that implies the PL condition for strongly convex functions: If $f$ satisfies the $\mu$-strong aiming condition, then

$$\langle \nabla f(x), \, x - \pi(x) \rangle \geq f(x) - f(\pi(x)) + \frac{\mu}{2} \|x - \pi(x)\|^2$$

and thus

$$\begin{aligned}
\|\nabla f(x)\| &\geq \left\langle \nabla f(x), \frac{x - \pi(x)}{\|x - \pi(x)\|} \right\rangle \\
&\geq \frac{f(x) - f(\pi(x))}{\|x - \pi(x)\|} + \frac{\mu}{2} \|x - \pi(x)\| \\
&\geq \min_{\xi > 0} \left( \frac{f(x) - f(\pi(x))}{\xi} + \frac{\mu}{2} \xi \right).
\end{aligned}$$

Setting the derivative with respect to $\xi$ to zero, we find that the minimum is achieved when

$$\frac{f(x) - f(\pi(x))}{\xi^2} = \frac{\mu}{2}, \quad \text{so } \xi = \sqrt{\frac{2\big(f(x) - \inf f\big)}{\mu}},$$

so

$$\|\nabla f(x)\| \geq \sqrt{\frac{\mu\big(f(x) - \inf f\big)}{2}} + \frac{\mu}{2} \sqrt{\frac{2}{\mu}} \big(f(x) - \inf f\big) = \sqrt{2\mu\big(f(x) - \inf f\big)}.$$

The PL condition follows by squaring both sides.

**Third claim.** Let $x_t = (1 - t)\pi(x) + tx$ for $0 \leq t \leq 1$. First we observe that $\pi(x_t) = \pi(x)$ for every $t \in [0, 1]$. Indeed, if there is another minimizer $z$ of $f$ such that $\|x_t - z\| \leq \|x_t, \pi(x)\|$ then

$$\|x - \pi(x)\| = \|x - x_t\| + \|x_t - \pi(x)\| \leq \|x - x_t\| + \|x_t - z\| \leq \|x - z\|$$

since $x, x_t$, and $\pi(x)$ all lie on a straight line. If there exists a unique closest point $\pi(x)$ in $\mathcal{M}$, then we find that $z = \pi(x)$.

We note that $x_t - \pi(x) = t(x - \pi(x))$ and thus

$$\frac{d}{dt} f(x_t) = \langle \nabla f(x_t), x - \pi(x) \rangle = \frac{1}{t} \langle \nabla f(x_t), \, x_t - \pi(x_t) \rangle \geq 0.$$

In particular, $t \mapsto f(x_t)$ is an increasing function on the set $I_x := \{t > 0 : x_t \in \mathcal{U}_\alpha\}$. If $I_x$ has multiple connected components in $(0, 1)$, this is only possible if $f = \alpha$ on the boundaries. If $f = \alpha$ on the lower boundary of a connected component, then $f \geq \alpha$ inside the entire interval as $f$ increases, contradicting the definition of $\mathcal{U}_\alpha$.

**Fourth claim.** Since $f$ is continuous, $\mathcal{U}_\alpha$ is open. Therefore, $x_t := x_t - t\nabla f(x) \in \mathcal{U}_\alpha$ if $t$ is small. If $f$ is $L$-Lipschitz continuous in $\mathcal{U}_\alpha$, then

$$f(x_t) \leq f(x) + \langle f(x), \, x_t - x \rangle + \frac{L}{2} \|x_t - x\|^2$$

for all $t$ such that $x_s \in \mathcal{U}_\alpha$ for $s \in (0, t)$ and $t \leq 1/L$ by Lemma 19. Since the function $t \mapsto f(x) - \frac{t}{2} \|\nabla f(x)\|^2$ is decreasing in $t$, we see that $x_t \in \mathcal{U}_\alpha$ for $t \in [0, 1/L]$. In particular, we note that

$$0 \leq f(x - \nabla f/L) - \inf f \leq f(x) - \frac{1}{2L} \|\nabla f(x)\|^2 - \inf f \leq f(x) - \frac{2\mu}{2L} (f(x) - \inf f) - \inf f$$

$$\leq (1 - \mu/L)(f(x) - \inf f),$$

implying the result.

**Fifth claim.** Assume that $x^*, x' \in \arg\min f$. Define $x_t = tx^* + (1 - t)x'$. Then

$$\frac{d}{dt} f(x_t) = \langle \nabla f(x_t), x^* - x' \rangle = \left\langle \nabla f(x_t), \frac{x_t - x'}{t} \right\rangle \geq \frac{\gamma}{t} \left( f(x_t) - \min f + \frac{\mu}{2} \|x_t - x'\|^2 \right) > 0$$

unless $\|x_t - x'\|^2 = 0$, i.e. unless $x^* = x'$. Thus $f(x_t)$ is strictly monotone increasing on $\{t \in [0, 1] : x_t \in U\}$. Since $x^* \in U$, this means that $f$ must be strictly increasing on the final segment $(1 - \xi, 1]$ where it reaches the global minimizer $x^*$ at $t = 1$, leading to a contradiction.

**Sixth claim.** This follows by essentially the same argument as the fifth claim: If $x$ is any point, then $tx + (1 - t)x^* \in U$ since $U$ is star-shaped about $x^*$ and

$$\frac{d}{dt} f(x_t) = \langle \nabla f(x_t), x - x^* \rangle = \left\langle \nabla f(x_t), \frac{x_t - x^*}{t} \right\rangle \geq \frac{\gamma}{t} (f(x_t) - f(x^*)) \geq 0.$$

In particular, $f$ is increasing along any rays starting at $x^*$, so if $f(x) < \alpha$, then $f(x_t) < \alpha$ for any $t \in [0, 1]$.

The set of minimizers is star-shaped about every minimizer, hence convex. $\qquad\square$

## C.2 A ONE-DIMENSIONAL EXAMPLE

In the following simple one-dimensional example, we illustrate the hierarchy of geometric conditions between convexity and the PL condition.

*Example* 25. Let $R > 0$ and $\varepsilon \in (0, 1)$. Consider the even function given for $x > 0$ by

$$f(x) = \frac{1 + \varepsilon \sin(2R \log x)}{2} x^2$$

$$f'(x) = \big(1 + \varepsilon \sin(2R \log x) + R\varepsilon \, \cos(2R \log x)\big) x$$

$$f''(x) = 1 + \varepsilon \sin(2R \log x) + R\varepsilon \, \cos(2R \log x) + 2R\varepsilon \, \cos(2R \log x) - 2R^2 \varepsilon \, \sin(2R \log x)$$

$$= 1 + \varepsilon(1 - 2R^2) \sin(2R \log x) + 3R\varepsilon \, \cos(2R \log x).$$

We first note that evidently $\frac{1-\varepsilon}{2} x^2 \leq f(x) \leq \frac{1+\varepsilon}{2} x^2$ for all $x$. In particular, $x^* = 0$ is the unique global minimizer of $f$.

**L-smoothness.** The function $g(\xi) = A\sin(\xi) + B\cos(\xi)$ attains its maximum when $A\cos\xi - B\sin\xi = 0$ for $A, B \in \mathbb{R}$, i.e. $\sin\xi = \frac{A}{\sqrt{A^2+B^2}}$ and $\cos\xi = \frac{B}{\sqrt{A^2+B^2}}$. In particular $\max_\xi g(\xi) = \sqrt{A^2 + B^2}$, so

$$|f''(x)| \leq 1 + \varepsilon\sqrt{\left(1 - 2R^2\right)^2 + (3R)^2} = 1 + \varepsilon\sqrt{1 + 5R^2 + 4R^4}.$$

The second derivative is discontinuous at $x^* = 0$, but bounded on $\mathbb{R} \setminus \{0\}$ by $L = 1 + \varepsilon\sqrt{1 + 5R^2 + 4R^4}$, i.e. $f$ is $L$-smooth.

**Convexity.** By the same consideration, we see that $f$ is convex if and only if $f'' \geq 0$, i.e. if and only if $\varepsilon\sqrt{1 + 5R^2 + 4R^4} \leq 1$. If the inequality is strict, $f$ is strongly convex with constant $\mu = 1 - \varepsilon\sqrt{1 + 5R^2 + 4R^4}$.

**PL inequality.** By the same argument as above, we see that

$$|f'(x)|^2 \geq \left(1 + \varepsilon\sin + R\varepsilon\,\cos\right)^2 x^2 \geq \left(1 - \varepsilon\sqrt{1 + R^2}\right)^2 x^2$$

if $\varepsilon\sqrt{1 + R^2} < 1$, where all trigonometric functions are evaluated at $\xi = 2R\log x$.

In particular, if $\varepsilon\sqrt{1 + R^2} < 1$, then

$$|f'(x)|^2 \geq \left(1 - \varepsilon\sqrt{1 + R^2}\right)^2 x^2 \geq 2\frac{\left(1 - \varepsilon\sqrt{1 + R^2}\right)^2}{1 + \varepsilon}\frac{1 + \varepsilon}{2}x^2 \geq 2\frac{\left(1 - \varepsilon\sqrt{1 + R^2}\right)^2}{1 + \varepsilon}f(x),$$

i.e. $f$ satisfies the PL condition with constant

$$\mu = \frac{(1 - \varepsilon\sqrt{1 + R^2})^2}{1 + \varepsilon}.$$

**Infinite number of local minimizers cluster at the orgin.** If $\varepsilon\sqrt{1 + R^2} > 1$ on the other hand, then by the same argument $f'$ changes sign an infinite number of times in any neighborhood of the origin since $\lim_{x\to 0^+}\log x = -\infty$.

**$\mu$-strong aiming condition.** Note that

$$x \cdot f'(x) - f(x) = \left(\frac{1}{2} + \frac{\varepsilon}{2}\sin(2R\log x) + R\varepsilon\,\cos(\log x)\right)x^2$$

$$\geq \left(\frac{1}{2} - \sqrt{(\varepsilon/2)^2 + (R\varepsilon)^2}\right)x^2$$

$$= \frac{1}{2}\left(1 - \varepsilon\sqrt{1 + 4R^2}\right)x^2.$$

In particular, $f$ is $\mu$-strongly aiming (with respect to the unique global minimizer) with

$$\mu = 1 - \varepsilon\sqrt{1 + 4R^2}$$

if $\varepsilon\sqrt{1 + 4R^2} < 1$ and it fails to be $\mu$-strongly aiming for any $\mu > 0$ otherwise.

**Quasar-convexity.** Since the minimizer is unique, $(\gamma, \mu)$-strong convexity is a strictly more general notion than strong aiming condition as we have an additional parameter $\gamma$ to relax the requirements. Essentially the same calculation reads

$$x \cdot f'(x) - \gamma f(x) = \left(1 - \frac{\gamma}{2} + \varepsilon\left((1 - \gamma/2)\sin(\dots) + R\cos(\dots)\right)\right)x^2$$

$$\geq \left(1 - \frac{\gamma}{2} - \varepsilon\sqrt{(1 - \gamma/2)^2 + R^2}\right)x^2.$$

In particular, $f$ is $\gamma$-quasar convex if

$$1 - \frac{\gamma}{2} - \varepsilon\sqrt{(1 - \gamma/2)^2 + R^2} \geq 0 \qquad \Leftrightarrow \qquad 1 - \varepsilon\sqrt{1 + \left(\frac{R}{1 - \gamma/2}\right)^2} \geq 0$$

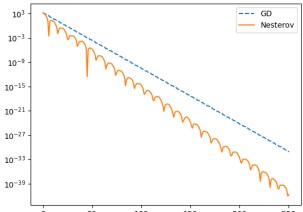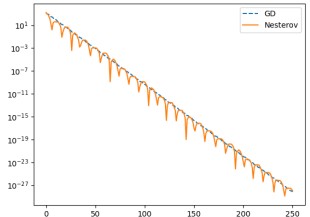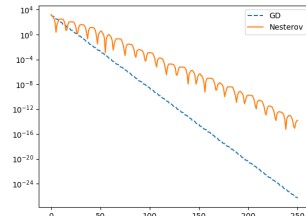

Figure 4: We compare the trajectories of gradient descent and Nesterov's algorithm for the objective function $f$ in Example 25 with $R = 6$ and $\varepsilon = 0.075$ (left), $\varepsilon = 0.08$ (middle) and $\varepsilon = 0.085$ (right). Evidently, if $\varepsilon\sqrt{1 + 4R^2}$ is very close to the threshold value 1, gradient descent outperforms Nesterov's algorithm with the theoretically guaranteed parameters.

$$\Leftrightarrow \quad 1 + \left(\frac{R}{1 - \gamma/2}\right)^2 \leq \frac{1}{\varepsilon^2}.$$

Such a $\gamma$ can be found if and only if $R^2 < \frac{1 - \varepsilon^2}{\varepsilon^2}$. Choosing $\gamma$ slightly smaller, it is then also always possible to make $f$ $(\gamma', \mu')$-strongly convex for some $\gamma' \in (0, \gamma)$ and $\mu' > 0$.

**Relationship between conditions.** We quickly summarize the various parameter ranges for which the function $f$ satisfies good geometric conditions.

|  | PL condition | $\mu$-strongly aiming | $\mu$-strongly convex |
|---|---|---|---|
| Must be $< 1$ | $\varepsilon\sqrt{1 + R^2}$ | $\varepsilon\sqrt{1 + 4R^2}$ | $\varepsilon\sqrt{1 + 5R^2 + 4R^4}$ |
| Constant | $\frac{(1 - \varepsilon\sqrt{1 + R^2})^2}{1 + \varepsilon}$ | $1 - \varepsilon\sqrt{1 + 4R^2}$ | $1 - \varepsilon\sqrt{1 + 5R^2 + 4R^4}$ |

Evidently, classical strong convexity implies strong aiming condition with respect to the global minimizer which in turn implies the PL condition. The parameter ranges and constants are generally vastly different. All estimates, except for the PL constant, are sharp. In particular, the one-dimensional examples demonstrate that strong aiming condition is strictly weaker than convexity along the line segment connecting $x$ to $\pi(x)$.

In the case of quadratic objective functions or functions modelled on them (such as the distance function from a manifold), the PL constant is essentially the same as the parameter of strong convexity. For the function $f$ in Example 25 on the other hand, the PL constant is noticeably larger than the parameter of strong convexity with respect to the unique minimizer. In Figure 4, we observe that with the parameters $\eta = 1/L$ and $\rho = (1 - \sqrt{\mu\eta})/(1 + \sqrt{\mu\eta})$, gradient descent may at times converge faster, at least with the parameter choice that is derived over a large function class rather than for an individual objective function.

### C.3 DEEP LEARNING

While some notions of a 'good' geometry are weaker than others (for instance, strong convexity implies the PL condition but not vice versa), they all share an important common feature: *If $x$ is a critical point of $f$, then it is a global minimizer.* Namely, the PL inequality implies that $\|\nabla f(x)\|^2 > 0$ unless $f(x) = \inf f$ and $\gamma$-quasar convexity implies that $\rangle\nabla f(x), x - x^*\langle > 0$ unless $f(x) = f(x^*) = \inf f$, showing that $\nabla f(x) \neq 0$.

In deep learning applications, critical points are guaranteed to occur under very general circumstances: If

$$f(\beta, a, W, b; x) = \beta + \sum_{i=1}^{n} a_i \sigma(w_i^T x + b)$$

is a neural network with a single hidden layer and a $C^1$-activation function satisfying $\sigma(0) = 0$ (e.g. $\tanh$), then the loss function

$$L(\beta, a, W, b) = \frac{1}{n} \sum_{j=1}^{n} \left| f(\beta, a, W, b; x_j) - y_j \right|^2,$$

satisfies

$$\nabla L(\beta, a, W, b) = 0$$

for $a = b = 0 \in \mathbb{R}^n$, $W = 0 \in \mathbb{R}^{n \times d}$ and $\beta = \frac{1}{n} \sum_{j=1}^{n} y_j$. The same is true if a row $w_i$ of $W$ is merely orthogonal to all data points $x_j$ but does not vanish.

For deeper networks, the set of critical points becomes larger: As long as the parameters of two layers are all zero, the remaining layers can be chosen arbitrarily. If more than two layers are all zero, then the also the second parameter derivative vanishes. In particular, the critical point for which all parameters are zero cannot be a strict saddle point.

While there are guarantees that individual algorithms escape certain types of critical points almost surely (e.g. strict saddles, (Lee et al., 2019; O'Neill & Wright, 2019)), they may take very long to do so (Du et al., 2017). The analysis of accelerated rates becomes asymptotic at best. We claim that our notion of strong aiming condition suffers from the same 'optimism' globally, but locally captures two important features of deep learning landscapes close to the set of global minimizers which are not captured by concepts which require a geometric condition with respect to *all* minimizers: A manifold along which we can move tangentially, and convexity in directions which are perpendicular to the manifold.

