# OpenReview forum: "Nesterov acceleration in benignly non-convex landscapes"
_ICLR.cc/2025/Conference — ICLR 2025 Spotlight_

### Official Review · Reviewer_cdrW · 2024-10-24

**Soundness:** 1
**Presentation:** 1
**Contribution:** 2
**Rating:** 6
**Confidence:** 4

**Summary:**

Motivated by nonconvexity of deep learning objectives, this paper studies the  heavy ball method under relaxed convexity assumptions. The continuous time, discrete time and stochastic algorithms are investigated. Some toy experiments are presented to illustrate how the convexity assumptions hold in practice.


------- AFTER REBUTTAL -------


**I raise my score to 6. The author(s) present interesting and sufficiently new contributions, but the paper needs strong revisions in terms of presentation and redaction.**

**Strengths:**

Benign convexity of machine learning problems is an interesting direction in order to understand why gradient methods may converge well in practice.

**Weaknesses:**

The author(s) confuse the assumptions used in the related work by Aujol et al., titled "Convergence Rates of the Heavy Ball Method for Quasi-Strongly Convex Optimization," published in SIAM Journal on Optimization in 2022.

In the introduction, the authors claim that "strong quasar-convexity" is used in Aujol et al. (2022). This condition appears to be stronger than the one used by the authors, referred to as Assumption 4, which the authors now describe as "strong convexity with respect to the closest minimizer."

**However, Aujol et al. (2022) already use Assumption 4.** Their assumption is not quasar-convexity, as mistakenly stated in the introduction, but rather strong convexity with respect to the projection onto the minimizing set, which corresponds to the closest minimizer (see equation (4) in Aujol et al., 2022). As a result, **all of the theoretical contributions presented by the authors already exist in the literature**. For example, Theorem 7 replicates the convergence rates from [Theorem 1, Aujol et al., 2022].

Furthermore, the authors conflate Nesterov's acceleration with the heavy ball method throughout the paper, although they are two distinct algorithms. **The paper focuses on the heavy ball method, not Nesterov's acceleration.**

**Questions:**

I suggest the authors to better compare their work with the related works.

---

> ### Author Response · Authors · 2024-11-18
>
> We thank the reviewer for their service. Unfortunately, **neither of their main claims and criticisms of our work is correct.**
>
> Aujol et al. (2022) make significantly stronger assumptions:
> 1. In the continuous time setting (Section 3), the authors assume that "$F:\mathbb R^n \to \mathbb R$ is a differentiable **convex** function admitting a **unique minimizer** $x^*$. Assume that $F$ is **additionally quasi-strongly convex**." (emphasis ours) All results in continuous time assume that $F$ has a *unique* minimizer. Our assumptions do not imply either convexity or uniqueness of minimizers, as also demonstrated by the examples in our Section 2.
> 2. In the discrete time setting (Section 4), they assume that $F$ is not just quasi-strongly convex, but in fact $\mu$-strongly convex. This, of course, does not imply any of the results proved in our work, where the focus is relaxing the assumptions on the objective function.
> 3. Aujol et al. do not consider the stochastic setting at all. A major part of our analysis is not considered in this work.
>
> Needless to say, this setting is significantly simpler and needs none of the innovations in our proofs which address tangential movement along the set of minimizers.
>
> These distinctions are not mere technicalities: In overparametrized machine learning, the minimizer is *never* unique, and deterministic optimization is rarely used. The claim that "all of the theoretical contributions presented by the authors already exist in the literature" is wildly inaccurate. **At the very least, the reference cited by the reviewer does not support their claim.** If there are other relevant works which we missed, we are grateful for further feedback.
>
> * We understand the reviewers confusion about Nesterov’s method and heavy ball momentum. Yurii Nesterov has introduced several algorithms in optimization theory. The algorithm we analyze is, in the deterministic setting, a reparametrization of the one introduced in "A Method for Solving the Convex Programming Problem with Convergence Rate $O(1/k^2)$" in 1982 (up to a different choice of the factor which in Nesterov’s original work is denoted by $(a_k-1)/ a_{k+1}$, which is standard in the strongly convex setting). The equivalence between the time stepping scheme that we use and the standard version of Nesterov's method is proved in Appendix B of the work of Gupta et al. (2023) cited in our bibliography.
>
>     The heavy ball discretization indeed looks very similar to this parametrization of Nesterov’s algorithm, but they are different. As noted by Sutskever et al. "On the importance of initialization and momentum in deep learning" (2013), the only difference between the two methods is the point at which the gradient is evaluated. While the heavy ball method uses gradients at $x_n$, Nesterov's method uses gradients evaluated at $x'_n = x_n + \rho v_n$. We make use of the latter in our work.
>
> We will be very happy to engage in further discussion, but we maintain that the reviewer's characterization of our contribution and comparison with *Convergence Rates of the Heavy Ball Method for Quasi-Strongly Convex Optimization* by Aujol et al. is highly inaccurate.

---

> > ### Comment · Reviewer_cdrW · 2024-11-21
> >
> > Thank you for your response. I apologize for the confusion.
> >
> > Indeed, Aujol et al. assume uniqueness of minimizers and their setting is stronger. However your literature review is really confusing. For instance Aujol et al. assume weak quasi convexity with respect to the closest minimizer and Guminov et al. assume weak quasi convexity to a fixed minimizer. In your introduction you claim these papers use the same hypothesis. This is inaccurate.
> >
> > I am not really convinced about the structure of your contributions. You introduce interesting technicalities on the geometry of minimizers, but the deep learning aspect is really incomplete. As I understand, you run 7 experiments on some minimizer $w$ (only one) to observe the convexity of the loss along the gradient direction, which can be highly imprecise given that your are close to the minimizing set. This has some illustrative value, but this is not sufficient to provide an experimental evidence.
> >
> >
> > I acknowledge the technical contribution on the proofs (and I apologize again for my wrong reading) but the overall presentation or the positioning has to be reworked, so I cannot go above the acceptance threshold.

---

> > > ### Author Response · Authors · 2024-11-30
> > >
> > > We are grateful to the reviewer for engaging in a constructive discussion. We acknowledge that the presentation and positioning are some important aspects of the paper. Taking the reviewer's feedback into account, we have uploaded a revision of the paper with an updated literature review, putting our contributions in context more clearly (also see the global response for details).
> > >
> > > >For instance Aujol et al. assume weak quasi convexity with respect to the closest minimizer and Guminov et al. assume weak quasi convexity to a fixed minimizer. In your introduction you claim these papers use the same hypothesis.
> > >
> > > We acknowledge that the two works use different hypotheses. The confusion was caused by the similar names used for some different properties across the literature.
> > >
> > > * Guminov et al. (2023) introduced their assumption by the name *weak quasi convexity* in 2017 (published 2023). Hinder et al. (2023) refer to the same condition by the name *quasar-convexity* (they justify this choice of renaming in a footnote in their paper), which is also the name that many of the subsequent papers use.
> > > * Aujol et al. (2022) use a different assumption called *quasi-strong convexity*, which was introduced by Necoara et al. (2019). The definition of quasi-strong convexity assumes strong convexity with respect to the closest minimizer. However, both Necoara et al. and Aujol et al. additionally assume that the objective function is convex and has a unique minimizer (thus effectively assuming strong convexity with respect to a fixed minimizer).
> > >
> > > We have further clarified these nuances in our updated literature review.
> > >
> > > We would kindly request the reviewer to take into consideration the improved presentation and to consider raising their score for the revised paper.

---

> ### Comment · Reviewer_cdrW · 2024-12-01
>
> Thank you for you effort for improving the paper. I mostly agree with your points from the general response:
>
> - You work essentially extends the literature on convergence guarantees of heavy ball/nesterov for beningly convex functions. You present techniques to improve existing results such as the one of Aujol et al., and to manage non-isolated minima under geometric assumptions.
>
> - The connection with deep learning is not central, and is mostly supported by the literature on overparameterized models (in particular in Cooper).
>
>
> I'm still not satisfied with the current redaction. It really has to be improved:
>
>
>
> - The introduction still need to be polished.  I believe the main elements which have to be exposed there are: (i) your motivations in deep learning; which is mainly the connection with overparameterized models, and (ii) a clear literature review of convergence guarantees under weak convexity assumptions (nonuniquess of minimizers etc). But the reading I have of your current introduction looks like this:
>
> > - Accelerated methods  in deep learning
> > - PL condition
> > - NTK
> > - Hessian eigen values
> > - Gradient method (with a reference to Cauchy)
> > - Nonsmooth optimization (with FISTA)
> > - Then, heavy ball in deep learning again
> > - KL inequality, with only one recent reference from 2023
> > - A small mention to time discretization but with no further explanation
> > - strong quasar convexity
> > - Implicit bias in deep learning
> > - Your contributions, with the first mention of "overparameterized models", and also this strange mention of the Hessian eigenvalues...
>
>
> - Sometimes your language is overblown:
>
> > In general deep learning applications, the set of minimizers of the loss function is a (generally curved)
> manifold
>
> "Deep learning applications" is wide... Can you claim this any model?
>
> > We have proved that first order momentum-based methods accelerate convergence in a **much** more
> general setting than convex optimization
>
> This sentence is too general while your work focuses on particular notions of weak convexity, and a specific landscape geometry.
>
>
>  > with many geometric features motivated by **realistic** loss landscapes in deep learning
>
> This is overblown. As you admit in your general comment, some works on overparameterized neural networks may motivate the use of such geometric features but the word "realistic" seems too much to me.
>
> - I would have preferred more insights into the proof techniques in the main paper. You mention that the main difficulty lies in controlling the "tangential motion" (I don't even know what it is precisely) but you do not explain how this is achieved or how it connects with the assumptions. I believe your proof techniques could contribute to the ML/optimization community from a theoretical perspective (more than your deep learning illustrations). However, the current presentation makes them difficult to access.
>
>
> **Overall, the paper contains new contributions for the convergence theory. Thus I raise my score to 6 but I urge the authors to work on the redaction for the future version.**

---

> > ### Author Response · Authors · 2024-12-02
> >
> > We thank the reviewer for their active engagement and their valuable suggestions! We will make sure to carefully revise the introduction to the final version of the article. Since the deadline for submitting a revised pdf has passed, we cannot update the paper right now, but we appreciate the points that the reviewer is making. We agree with the reviewer's assessment that these are not major (but still useful) changes and can be easily incorporated into the final version, which we will make sure to do.
> >
> > > "Deep learning applications" is wide... Can you claim this any model?
> >
> > To be more precise, it holds for sufficiently smooth overparametrized models that can fit the training data exactly. This covers a large and important class of problems, but certainly not the entirety of deep learning applications. We will be more precise here, and we will be more judicious in the use of words like ‘realistic’ elsewhere in the paper as well.
> >
> > > I would have preferred more insights into the proof techniques in the main paper.
> >
> > This is a good suggestion. We agree that including a discussion on the proof techniques or a sketch of the proof will be useful and give more insight into our work. We can move some of the example figures to the appendix in order to include such a discussion/sketch in the main body.

---

### Official Review · Reviewer_4krH · 2024-11-01

**Soundness:** 2
**Presentation:** 3
**Contribution:** 2
**Rating:** 5
**Confidence:** 3

**Summary:**

The paper presents convergence analysis of Nesterov Acceleration for nonconvex functions. Some (geometric) assumptions weaker than PL inequality are employed to derive convergence results for NAG.

**Strengths:**

There are some works have been reviewed in the literature, both from the line of momentum algorithms, and the geometry of deep neural networks. The paper is well-organized.

**Weaknesses:**

1. To the best of my knowledge, both from theoretical and practical perspectives, the momentum methods commonly used in deep learning are based on the Polyak Heavy-ball method, not Nesterov acceleration. However, the paper under review focuses more on Nesterov acceleration, while mentioning motivation from deep learning without any references. Could the author carefully provide sufficient references to support this motivation?

2. The main contributions of the paper in comparison with related references is unclear. Could you state them more explicitly, discuss them in greater detail?

3. The paper lacks a thorough literature review. For example, a unified convergence analysis for momentum algorithms has already been considered in Josz et al. (https://epubs.siam.org/doi/abs/10.1137/23M1545720?journalCode=sjope8). Could the authors please address this paper carefully and consider the references therein to better position their contribution within the literature on Nesterov acceleration?

4. The assumptions in Section 2.1 seem too strong, as they imply the PL inequality. Recent works, such as Josz et al. mentioned above, require only the KL inequality, which holds for a significantly larger class of functions encountered frequently.

5. What is the connection between your continuous-time and discrete-time analyses? While the discrete-time analysis is of primary interest, the purpose of including the continuous-time analysis is unclear, aside from serving as additional motivation. If the continuous-time analysis can be directly applied to derive results for discrete time, its inclusion is entirely appropriate. However, if it is independent of the discrete-time analysis, I do not think it should be included unless carefully justified.

6. The numerical experiments are too simple. Could you include additional experiments on practical classes of nonconvex functions, such as those commonly found in deep learning?

**Questions:**

See weakness

---

> ### Author Response · Authors · 2024-11-18
>
> > To the best of my knowledge, both from theoretical and practical perspectives, the momentum methods commonly used in deep learning are based on the Polyak Heavy-ball method, not Nesterov acceleration. […] Could the author carefully provide sufficient references to support this motivation?
>
> Theoretical guarantees for heavy ball momentum are very weak compared to Nesterov momentum, even in the setting of convex optimization. Popular machine learning libraries like PyTorch and TensorFlow include easy options to to use either Nesterov or heavy ball momentum in their SGD optimizer. In *On the importance of initialization and momentum in deep learning* (ICML 2013), Sutskever et al. study both heavy ball and Nesterov momentum and highlight situations where latter is more effective than the former. Other works suggest that using Nesterov momentum with adaptive optimizers may indeed lead to superior performance (e.g. Dozat: *Incorporating Nesterov Momentum into Adam* (2016), Xie et al: *Adan: Adaptive Nesterov Momentum Algorithm for Faster Optimizing Deep Models* (2024)).
>
> We take a broad view of the role of optimization theory for machine learning, where we consider not only the currently dominant paradigm, but also emerging techniques which may evolve into dominant narratives in the future.
>
> > The main contributions of the paper in comparison with related references is unclear. Could you state them more explicitly, discuss them in greater detail?
>
> We will be happy to expand the literature review in a revised version. The main novelty of our analysis is the extension to a setting where minimizers are not unique and where movement tangential to the set of minimizers has to be carefully considered (as is the case in overparametrized deep learning).
>
> > For example, a unified convergence analysis for momentum algorithms has already been considered in Josz et al.
>
> We are grateful to the reviewers for pointing us towards this reference. We will include in in the literature review in a revised version of the article. To us, the main differences between the articles are:
>
> * We make more restrictive geometric assumptions, but obtain a stronger result (see also the next point).
>
> * We include an analysis of a broad stochastic setting with state-dependent noise. Stochastic variations of the analysis are listed as future directions in the work of Josz et al.
>
> > Recent works, such as Josz et al. mentioned above, require only the KL inequality, which holds for a significantly larger class of functions encountered frequently.
>
> Josz et al. prove convergence to a local minimizer under certain mild regularity conditions. We make stronger geometric assumptions and obtain stronger quantitative guarantees on the speed of convergence. In particular, we obtain an ‘accelerated’ rate of convergence. As Yue et al. (2023, details in bibliography) prove, no such guarantee can be found under weaker assumptions such as the PL condition.
>
> > If the continuous-time analysis can be directly applied to derive results for discrete time, its inclusion is entirely appropriate.
>
> The proofs in discrete time are more technical variations of proofs in continuous time. We include them to provide geometric intuition and illustrate the important ideas in a simpler setting. Using continuous time differential equations to gain insight into the behavior of and develop methods of analysis for discrete time algorithms is well-established and has led to tremendous progress following e.g. the highly influential work of Su et al. *A Differential Equation for Modeling Nesterov's Accelerated Gradient Method: Theory and Insights* (NeurIPS 2014, JMLR 2016).
>
> Additionally, the results obtained in continuous time are stronger in terms of the geometry. In the current work, we decided to extend the results towards the stochastic direction rather than full geometric generality. Nevertheless, we believe that the continuous time analysis illustrates other directions in which our results can presumably be extended in future work.
>
>  > The numerical experiments are too simple
>
> The goal of the current work is to propose an appropriate setting for the analysis of certain momentum-based optimization algorithms that are already being used in deep learning sucessfully, albeit without convergence guarantees. We do not propose any new algorithms but provide theoretical guarantees for existing ones with significantly weaker assumptions. Numerical experiments would mostly recreate well-known results and were not deemed a priority. This is particularly true for Nesterov’s algorithm, which has been used for neural network training for over a decade.
>
> In the AGNES setting, admittedly, numerical studies are scarcer, and the image classification experiments of Gupta et al. do not match theoretical assumptions precisely. We would be happy to include further experiments in different settings where MSE loss is common, such as regression or the training of neural PDE solvers.

---

> > ### Comment · Reviewer_4krH · 2024-11-25
> >
> > Thanks for the clarifications provided by the authors. Unfortunately, I cannot increase the score, as your responses did not convincingly demonstrate the contributions of the paper. Below are some further comments, I hope can help to improve your paper, with point 4 being the major concern:
> >
> > 1. It is subjective to claim that "Theoretical guarantees for heavy ball momentum are very weak compared to Nesterov momentum, even in the setting of convex optimization," especially since the authors have not thoroughly reviewed the literature, including at least the work by Josz et al. and references therein.
> >
> > 2. I agree that there are scenarios where using Nesterov momentum is advantageous. However, your evidence does not convincingly show that Nesterov momentum is more commonly used than the heavy ball method. That said, I believe deriving the convergence properties for Nesterov momentum is necessary, and your evidence does support this claim effectively.
> >
> > 3. I attempted to review the paper "On the Importance of Initialization and Momentum in Deep Learning" but could not locate the evidence supporting your claim that "it highlights situations where the latter is more effective than the former." Could you please point me to the relevant sections for better understanding?
> >
> > 4. I believe it is necessary to delve deeper into the literature, such as the paper by Josz et al. and its references.  For example, since the analysis of the paper is based on KL property, the derivation convergence rates for the algorithms are rather simple and thus usually neglected in most recent studies. Additionally, stochastic variants are addressed in "Convergence of SGD with Momentum in the Nonconvex Case: A Novel Time Window-Based Analysis" by Junwen Qiu, Bohao Ma, and Andre Milzarek, where various types of convergence are considered.
> >
> > 5. From your response, it seems that the continuous-time analysis does not directly impact discrete-time results. While I have a different perspective on whether continuous-time analysis should be included, I respect your explanation on this matter.
> >
> > 6. Regarding numerical experiments, I fully agree that this is a minor point, and it is not necessary to improve them in the current version of the paper. However, if you come up with new ideas leading to additional numerical experiments, it would undoubtedly strengthen the paper.

---

> > > ### Author Response · Authors · 2024-11-30
> > >
> > > > It is subjective to claim that "Theoretical guarantees for heavy ball momentum are very weak compared to Nesterov momentum, even in the setting of convex optimization," especially since the authors have not thoroughly reviewed the literature, including at least the work by Josz et al. and references therein.
> > >
> > > This is incorrect — there are well-established results by Lessard et al. *Analysis and Design of Optimization Algorithms via Integral Quadratic Constraints* (2016) and Goujaud et al. *Provable non-accelerations of the heavy-ball method* (2023) which show that the heavy ball method may fail to converge at an accelerated rate or even fail to converge entirely in settings where Nesterov’s method remains stable. We have included these references in our revised literature review.
> > >
> > > > I attempted to review the paper "On the Importance of Initialization and Momentum in Deep Learning" but could not locate the evidence supporting your claim that "it highlights situations where the latter is more effective than the former." Could you please point me to the relevant sections for better understanding?
> > >
> > > We were referring to the section 2.1 (The Relationship between CM and NAG) in that paper. The section provides some heuristics about the advantages of NAG over heavy ball momentum as well as some rigorous justifications for the same in the case of quadratic objective functions.
> > >
> > > As we understand it, the reviewer's primary concerns are about better placing our contributions in the context of the existing literature. We acknowledge that it is an important aspect of the paper and thank the reviewer for pointing that out. We have uploaded a revision of our paper with an expanded literature review, more clearly delineating our own contributions. We have included the work of Josz et al. in the literature review, but we maintain that it is only tangentially relevant: We focus on accelerated rates of convergence. In the more general setting of Josz et al., these *cannot* be established as proved by Yue et al. *On the Lower Bound of Minimizing Polyak-Łojasiewicz Functions* (2023).
> > >
> > > We would kindly request the reviewer to consider raising their score for the revised version of the paper.

---

> > > > ### Comment · Reviewer_4krH · 2024-12-01
> > > >
> > > > Thank you for your clarifications. While not all of my concerns have been resolved, I am happy to increase my score based on the improvements made by the authors.

---

> ### Comment · Area_Chair_AbmS · 2024-11-25
>
> Dear Reviewer 4krH,
>
> The author discussion phase will be ending soon. The authors have provided detailed responses. Could you please reply to the authors with whether they have addressed your concern and whether you will keep or modify your assessment on this submission?
>
> Thanks.

---

### Official Review · Reviewer_jpMm · 2024-11-04

**Soundness:** 3
**Presentation:** 4
**Contribution:** 3
**Rating:** 8
**Confidence:** 3

**Summary:**

This paper studies the convergence rate of momentum-based optimization methods under more general conditions than (strong) convexity. Motivated by existing works, the authors assume a localized version of strong convexity that allows for multiple local minimizers. The authors provide examples of such objective functions and also show empirically that the assumption is reasonable in deep learning applications. By considering a continuous-time ODE, the authors characterize the behavior of momentum-based method and prove accelerated rate compared with standard GD. The setting of stochastic optimization is also considered as an extension.

**Strengths:**

1. Overall I think this is a nice paper with interesting contributions to the optimization community. I like the writing style of this paper, especially for the concise introduction and discussions of relevant works.

2. This paper justifies the superiority of momentum methods under weaker assumptions than strong convexity. Moreover, the assumptions made in this paper are supported by empirical evidence.

**Weaknesses:**

While I do not check all the proofs, the paper does not seem to have many novel technical contributions compared with existing works on ODE modeling of optimization algorithms. I'm not sure if this should be considered as a weakness, since the results themselves are interesting.

**Questions:**

I do not have any questions for this work.

---

> ### Author Response · Authors · 2024-11-18
>
> We thank the reviewer for their kind and helpful feedback.
>
> > the paper does not seem to have many novel technical contributions compared with existing works on ODE modeling of optimization algorithms
>
> There are parts of the article where new ideas are required for the ODE section (Lemma 1, and in the proof of the global convergence result of Lemma 8), but we agree that the main innovation is based on novel geometric insight, not the technical tools used for the analysis. The proofs in discrete time (Theorems 11, 13, 14 and 15) also build on top of existing proof ideas, but are more challenging. They require a much more careful analysis of terms because we need to handle the tangential and normal components of the velocity and the gradients separately.

---

> > ### Comment · Reviewer_jpMm · 2024-11-25
> >
> > Thank you for your reply. I will maintain my score and recommend acceptance of this paper.
> >
> > As other reviewers pointed out, the applications to deep learning might be a little bit confusing, and I would suggest the authors to modify that part in a revised version. However, considering the fact that this is largely a theoretical paper, experiments may not be the most crucial part. Also in modern deep learning, it seems that one can never rigorously justify the assumptions made for analyzing an optimization problem; instead I think it would be more important for the assumptions to reflect the practical challenges to some extent.
> >
> > For the current paper, I agree with other reviewers that more clarifications about assumptions need to be addressed more carefully. However, to the best of my knowledge, these assumptions are quite different from existing ones. for instance, the standard quasar convexity assumption requires uniqueness of minimizers, which does not hold in deep learning applications. On the other hand, the weaker PL condition excludes the possibility of acceleration as previously shown in the literature. The authors appear to figure out a set of assumptions that achieve a tradeoff between practical considerations and theoretical elegence. As a result I believe that this paper would be a nice contribution to the non-convex optimization community.

---

### Official Review · Reviewer_jZe8 · 2024-11-05

**Soundness:** 4
**Presentation:** 4
**Contribution:** 3
**Rating:** 8
**Confidence:** 3

**Summary:**

This paper presents an analysis of NAG that proves accelerated convergence rates in some non-convex scenarios, using assumptions on the loss function inspired by overparameterized deep learning. The authors analyze continuous time with deterministic gradients, discrete time with deterministic gradients, and discrete time with stochastic gradients (and multiplicative noise), showing accelerated convergence rate (better dependence on condition number) compared to gradient descent/gradient flow.

**Strengths:**

1. The paper is very clearly written. Previous work is totally explained and the authors are very straightforward when describing their novel contribution (Lines 42-51 and Lines 106-107).
2. The convergence rates are a clear improvement upon previous work, which either achieves similar rates with stronger assumptions or achieves a non-accelerated rate for gradient flow under a similar setting as this work (aiming condition).
3. The authors are straightforward about the limitations of their work, which I appreciate. In Lines 253-260, they mention that their global assumption on the loss geometry can only be guaranteed locally. Still, it seems that the assumptions are weaker than previous works (e.g. quasar convexity), so I think that this is acceptable.

I want to mention to other reviewers/AC that I am not too familiar with this particular line of work, so I am slightly less confident about the technical contribution of this submission. However, according to my best judgement, the contribution seems worthy of publication.

**Weaknesses:**

1. The assumptions are inspired by overparameterized deep learning, but are unlikely to be completely accurate. This authors already comment on one instance of this (Lines 253-260). Another instance is the requirement that the objective is $C^1$-smooth (Line 114), which is not satisfied by non-smooth activation functions. Still, the assumptions are weaker than previous work, and I believe these gaps are very non-trivial to address. This is a minor weakness compared to the strengths of the paper.

**Questions:**

1. I'm not sure to what degree the proof of Theorem 7 requires novel techniques. The authors mention (Line 324) that their overall proof strategy is common, and the contribution comes from their Lemma 1. I am not too familiar with this line of work, so it is hard for me to judge the novelty and non-triviality of Lemma 1. Can you elaborate on the technical contribution of Lemma 1?
2. Do you believe that similar ideas could be used for non-smooth objectives in future work? For example, the objective function for training a ReLU network with squared loss is almost everywhere continuously differentiable, instead of continuously differentiable (as required by your assumption #1). Could this gap be addressed with the kind of techniques used in this paper, or might this setting require entirely an entirely different approach?

---

> ### Author Response · Authors · 2024-11-18
>
> We thank the reviewer for their kind and helpful feedback.
>
> > I'm not sure to what degree the proof of Theorem 7 requires novel techniques.
>
> The proof of Theorem 7 combines two main points:
>
> * The trajectory of $x(t)$ remains in a good neighborhood of minimizers. This is achieved by energy estimates.
> * A modified total energy is decreasing. This mimics existing proofs using Lyapunov functions, but requires greater care since a 'base point' in the construction of the Lyapunov function is moving. Lemma 1 is needed to address this.
>
> As such, the proof of Theorem 7 does not require the development of novel tools so much as new geometric insight to combine known arguments. The proofs of Theorems 11, 13, 14 and 15 (i.e. in the discrete time setting ) are more challenging. They require a much more careful analysis of terms because we need to handle the tangential and normal components of the velocity and the gradients separately.
>
> > Do you believe that similar ideas could be used for non-smooth objectives in future work
>
> The reviewer raises an interesting question. We require control of the Lipschitz constant of the gradient in two places:
>
> 1. To bound the error in a gradient descent step. For non-smooth convex optimization, it is in places possible to circumvent this control at the cost of slower rates of convergence or without deterioration by selecting a partially implicit time stepping scheme (like FISTA, which treats part of the gradient implicitly). Intuitively, we see no obstructions to mimicking either one.
>
> 2. To track how the objective function $f$ changes in the momentum step. Here, only a lower bound on the eigenvalues of $D^2 f$ is required, and an assumption of the nature that $f(x) + \frac {\epsilon}2 \|x\|^2$ is convex may be sufficient here. We do not believe that a geometric assumption of this form can be removed entirely, but smoothness may not be needed.
>
> For a neural network with a single hidden layer $f_{(a,W)} (x) = \sum_{i=1}^n a_i\sigma(w_i^Tx)$ and the loss function $L(a, W) = | f_{(a,W)}(x) - y|^2$ with a single given data/label pair $(x,y)$, we find that $\nabla_{W} L = 2 (f_{(a,W)}(x) - y) \nabla_W f_{(a,W)}(x)$ and $D^2_W L = 2 (f_{(a,W)}(x) - y) D^2_W f_{(a,W)}(x) + 2 \nabla_W f_{(a,W)}(x) \otimes \nabla_W f_{(a,W)}(x)$.
>
> The second term in the sum is a non-negative semi-definite symmetric rank one tensor and therefore unproblematic. Since $f$ is non-linear in $W$, the Hessian $D^2_Wf$ is non-zero (a measure, if $\sigma$ is ReLU), and in general, the first term may be indefinite. If we choose $y$ adversarially, we can guarantee that $D^2L$ has a large negative eigenvalue even for smooth activation. If $\sigma$ is ReLU, then $D^2_Wf$ is singular with respect to Lebesgue measure, so intuitively there will be an 'infinitely large negative eigenvalue' whenever $f(x)-y$  has the wrong sign, even in a neighborhood of the set of global minimizers.
>
> Whether $L$ can be controlled locally in a sufficient manner, possibly under additional geometric assumptions, is an interesting question even in the shallow case, but beyond the scope of this work. We strongly believe that the results are relevant in a more general setting than we can guarantee at this point, and we believe that charting the borders of their validity is an exciting topic for future research.

---

> > ### Comment · Reviewer_jZe8 · 2024-11-20
> >
> > Thank you for your informative response. I agree that similar results can perhaps be obtained in more general settings, and I look forward to seeing this line of work develop. I am satisfied to keep my score.

---

### Author Response · Authors · 2024-11-30

We thank all reviewers for their helpful feedback and for actively engaging in the author discussion period.

All reviewers agree that there are novel contributions in our work. Their main criticism is that we did not explain clearly enough what these contributions are in the original submission, and did not outline clearly enough differences from the existing literature. We address these criticisms in a revised submission (changes marked in blue in the document). We would also like to highlight that we made compare our assumptions and other notions like quasar-convexity, strong convexity, and the PL condition in Lemma 24 in Appendix C.

Additionally, some reviewers note that we do not support our geometric modelling choices with extensive numerical experiments. In response, we hold that

1. Our assumptions are well-motivated from theoretical predictions, and a similar framework has been studied e.g. by Rebjock and Boumal (2023). The simulations are merely meant as an illustrative proof of concept and should not be considered an integral part of this work. Our theoretical contributions stand on their own.

2. It is well-known that the minimizers of objective functions in over-parametrized learning are non-isolated, and that Hessian eigenvalues may be negative, even close to global minimizers. Our analysis is the first to address objective functions with non-isolated minimizers in a meaningful way for momentum-based optimization. We also give a precise quantification of the impact of the magnitude of negative eigenvalues by the parameter $\varepsilon>0$.

	Whether or not the simulations suggest that this is a fully realistic setting for optimization in deep learning, we claim that it is undebatably a significant step towards a realistic model. We conjecture that it remains simplified, but that it captures important features much more precisely than previous assumptions.

3. Finally, we are the first to analyze the impact of negative eigenvalues of the Hessian for accelerated rates in the context of *stochastic* optimization. Namely, we show that these are generally *not* the limiting factor on achieving fast convergence if the gradient oracle is reasonably noisy (in a precise way).

Our links to deep learning are heuristic, but not novel — indeed, it can be argued that the study of accelerated first order methods is largely motivated by the fact that these methods are highly popular in deep learning. The main contribution of our work is not on establishing such a connection, but on extending guarantees in optimization theory to a setting which we merely argue to be *closer* to deep learning than previous work in this direction. To emphasize that, we have also renamed Section 2.3 to "Connection to Deep Learning" in order to avoid giving the impression that we are claiming to offer new applications of these already existing methods.

---

### Meta-Review · Area_Chair_AbmS · 2024-12-20

**Metareview:**

This paper investigates momentum-based optimization algorithms, such as Nesterov's accelerated gradient descent (NAG), under relaxed convexity assumptions inspired by the non-convexity of deep learning objectives. The analysis encompasses continuous-time models with deterministic gradients, discrete-time models with both deterministic and stochastic gradients, and cases where the noise exhibits additive and multiplicative scaling. These findings bridge the gap between theory and practice by extending momentum-based optimization guarantees to optimization problems with benign non-convexity.

The convergence rates achieved in this paper are an improvement over previous work, either matching rates under weaker assumptions or outperforming non-accelerated rates in similar settings. This paper justifies the superiority of momentum methods under relaxed convexity assumptions, which are supported by empirical evidence, and explores the intriguing concept of benign convexity to explain the practical effectiveness. These contributions represent a valuable addition to the optimization community.

The paper has several weaknesses, though many are minor compared to its strengths. The assumptions made may not fully reflect practical scenarios inspired by overparameterized deep learning. For instance, the requirement of smoothness excludes non-smooth activation functions, and the assumptions in Section 2.1 are criticized as overly strong, implying the PL inequality, whereas weaker conditions like the KL inequality are more broadly applicable. Additionally, the focus on Nesterov acceleration is questioned by one reviewer, as momentum methods commonly used in deep learning are often based on the heavy-ball method, and the paper lacks sufficient references to support its motivations. Also, the relationship between the continuous-time and discrete-time analyses is unclear, raising questions about the necessity of including the former unless its connection to the latter is better established.

**Additional Comments On Reviewer Discussion:**

Reviewer jZe8 pointed out that the smoothness assumption does not hold for some activation function. The authors did not fully addressed this point but I think it just the scope of this work, which is reasonable for a theoretical paper, so this is a minor issue in my opinion.

Reviewer jZe8 and Reviewer jpMm questioned the novelty of the theoretical proofs. The authors pointed out where the novelty comes and the reviewers are satisfied.

Reviewer 4krH was concerned that the NAG method is not as popular as the heavy-ball method for training deep learning models. He/she also pointed that the connection between the continuous-time and discrete-time analysis is not clear. Both questions are reasonable. I agree with the reviewer but I also feel that understating  the NAG method is also interesting for optimization community. I suggested the authors better motivate the necessity of studying the continuous time convergence. Reviewer 4krH  raised the score after rebuttal.

Reviewer cdrW had some confusions but the authors have clarified them. Reviewer cdrW then raised the score although he/she still believed the presentation and positioning of this paper should be improved.

---

### Decision · Program_Chairs · 2025-01-22

Accept (Spotlight)